# On the Limitation and Redundancy of Transformers: A Rank Perspective

## Abstract

Transformers have showcased superior performances across a variety of real-world applications, particularly leading to unparalleled successes of large "foundation" models. However, since these models are usually trained on web-scale datasets, the overall computation and memory loads are considerably increasing, calling for more *efficient* methods in machine learning. In this work, we step towards this direction by exploring the architectural limitation and redundancy of Transformers via investigating the ranks of attention score matrices. On one hand, extensive experiments are conducted on various model configurations (model dimensions, heads, layers, etc) and data distributions (both synthetic and real-world datasets with varied sequence lengths), uncovering two key properties: although the attention rank increases with the head dimension $d_h$, as expected, the rank is eventually upper bounded (limitation) and gets saturated (redundancy). We call them the *low-rank barrier* and *model-reduction effect*, respectively. On the other hand, we provide rigorous demonstrations for these observations through a fine-grained mathematical analysis, highlighting (i) a consistent theoretical upper bound ($\approx 0.63n$, $n$: the sequence length) of the attention rank regardless of the head dimension $d_h$, and (ii) a critical position of the rank saturation ($d_h = \Omega(\log n)$). These results shed light on the inductive biases and internal dynamics of Transformers, contributing to the theoretical understanding and assessment of the model capacity and efficiency in practical applications.

## 1 Introduction

In recent years, Transformer-based neural network models have reshaped the landscape of machine learning, demonstrating unparalleled successes across a myriad of applications including natural language processing (NLP) (Vaswani et al., 2017; Devlin et al., 2019; Raffel et al., 2020; Radford et al., 2018; Rae et al., 2021; Dehghani et al., 2023; Touvron et al., 2023; Liu et al., 2019; Hao et al., 2020; Liu et al., 2021; Yuan et al., 2022), computer vision (CV) (Chen et al., 2021b; Wang et al., 2022; Liang et al., 2021; Lu et al., 2022; Zhu et al., 2021; Wang et al., 2021), audios (Sung et al., 2022; Tsimpoukelli et al., 2021; Li et al., 2022), interdisciplinary sciences (Jumper et al., 2021), and so on. The core architecture module, anchored by the so-called attention mechanism, has been proved as a cornerstone particularly in capturing relationships with intricacies and nuances.

Mathematically, the central attention mechanism is designed to weigh the significance and correlations of input sequences via, e.g. inner products between trainable transformations on inputs (e.g. tokens), which is formulated as the attention score matrices. As a fundamental algebra concept, the matrix rank is supposed to impact the capacity (expressive ability) and learning performance of the attention mechanism and hence Transformer models. Particularly, an important phenomenon called the *low-rank bottleneck* is uncovered by numerous recent works (Kanai et al., 2018; Bhojanapalli et al., 2020; Dong et al., 2021; Lin et al., 2022), and several Transformer-based variants aim to reduce the computational and memory bottlenecks of modeling long sequences from the perspective of attention ranks (Chen et al., 2021a; Wang et al., 2020; Hu et al., 2022; Guo et al., 2019; Lin et al., 2022). However, these studies in general (i) are insufficient to quantitatively characterize the attention rank's *limitation* (i.e. low-rank upper bounds); (ii) lack theoretical analysis of the attention rank's *redundancy* (i.e. model-reduction). Based on (i), (ii) is straightforwardly applicable in practice, particularly in the current era of "foundation" models, where the pre-training efficiency on notable large models and web-scale datasets turns out a remarkable problem.

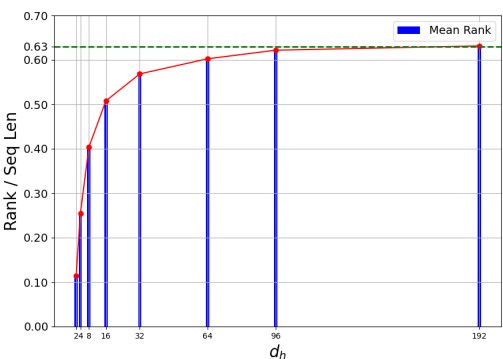

Figure 1: A typical phenomenon of the attention rank of an initialized Transformer model for different head dimensions $d_h$. Here, we evaluate a standard one-layer Transformer encoder block with $d_{\text{model}} = 384$ and the feed-forward hidden dimension of 512. We select $d_h \in \{2, 4, 8, 16, 32, 64, 96, 192\}$. The model weights are i.i.d. initialized using a standard normal distribution $\mathcal{N}(0, 1)$. The entries of input sequences are also independent $\mathcal{N}(0, 1)$ random variables, with a shape of $(n, b, d)$, where the sequence length $n$ is 100, the batch size $b$ is 32 and the data dimension $d = d_{\text{model}} = 384$. See details in Section 3.1.

In this work, we make an initial step towards this direction by studying the limitation and redundancy of general Transformers from the perspective of attention ranks. Figure 1 shows a typical experimental observation in the present work, focusing on the variation of attention ranks with respect to the pivotal head dimension ($d_h$). We observe that: (i) The attention rank increases with the head dimension. As $d_h$ increases within relatively small values, the increment of attention ranks is significant; (ii) For appropriately large values of $d_h$, further increases in $d_h$ lead to a diminishing return in the enhancement of attention ranks, with an ultimate upper bound of approximately $0.63n$, which is away from the full rank $n$ ($n$: sequence length and attention matrix size). Extensive experiments are performed, which consistently demonstrate these observations across various model and data settings, including varied model dimensions, different heads and layers, a variety of data distributions with increasing sequence lengths for both synthetic and real-world datasets. Theoretically, a fine-grained mathematical analysis is provided to rigorously support these experimental observations in a quantitative manner, including that (i) the attention rank has a consistent theoretical upper bound ($\approx 0.63n$) for any $d_h$, which shows the existence of the low-rank barrier ($n$ is the full-rank); (ii) when $d_h = \Omega(\log n)$, the attention rank gets saturated in the sense that further increasing the head dimension leads to diminishing rank enhancement. This study focuses on the model biases inherently in Transformer models, and the developed results not only shed light on the internal dynamics of Transformers, but also provide new insights to evaluate the model capacity and efficiency.

Our main contributions are summarized as follows:

1. Empirically, under extensive settings for the most general Transformer models and real-world datasets, it is shown that as the head dimension $d_h$ increases, the attention rank rises as expected, but the increment slows down significantly and eventually gets saturated, without reaching the full-rank (for appropriately large $d_h$).

2. Theoretically, mathematical estimates are established on the barrier of attention ranks, with an upper bound of approximately $0.63n$ (aligned with experimental observations). Moreover, after the critical position $d_h = \Omega(\log n)$ (also numerically verified), the attention rank gets saturated with negligible increments even by significantly increasing the head dimension.

The rest of this paper is organized as follows. In Section 2, we formulate the problem by reviewing the common Transformer architecture with the multi-head attention mechanism. Section 3 provides fundamental observations with various experiments and ablation studies. Section 4 includes the fine-grained mathematical analysis on the attention rank. Section 5 further verifies the developed results

on real-world datasets. Discussions on the related work, all the details of proofs and supplementary experiments can be found in the appendix.

**Notations** Throughout this paper, we use normal letters to denote scalars. Boldfaced lower-case/capital letters are reserved for vectors/matrices. Let $[n] := \{1, 2, \ldots, n\}$ for $n \in \mathbb{N}_+$. Let $\|\mathbf{x}\|_p := \left(\sum_{i=1}^n |x_i|^p\right)^{1/p}$ be the $\ell^p$-norm for $\mathbf{x} \in \mathbb{R}^n$ and $p \in [1, \infty]$, and $\|\mathbf{A}\|_F := \left(\sum_{i=1}^m \sum_{j=1}^n a_{ij}^2\right)^{1/2}$ be the Frobenius norm for $\mathbf{A} \in \mathbb{R}^{m \times n}$. Denote the standard basis of $\mathbb{R}^n$ by $\{\mathbf{e}_i\}_{i=1}^n$, i.e., $\mathbf{e}_i$ is the vector of all zeros except that the $i$-th position is 1. Let $\mathbf{0}_n \in \mathbb{R}^n$ be the vector of all zeros. For a probability space $(\Omega, \mathcal{F}, \mathbb{P})$, the probability of a measurable event $E \in \mathcal{F}$ is $\mathbb{P}(E)$. Let $\mathcal{N}(\boldsymbol{\mu}, \boldsymbol{\Sigma})$ be the multivariate normal distribution defined on $\mathbb{R}^n$, where $\boldsymbol{\mu} \in \mathbb{R}^n$ is the expectation and $\boldsymbol{\Sigma} \in \mathbb{R}^{n \times n}$ is the covariance. We use the big-O/big-Omega notation $f(n) = O(g(n))/f(n) = \Omega(g(n))$ to represent that $f$ is bounded above/below by $g$ asymptotically, i.e., there exists $c > 0, n_0 \in \mathbb{N}_+$ such that $f(n) \leq cg(n)/f(n) \geq cg(n)$ for any $n \geq n_0$.

## 2 PROBLEM FORMULATION

Consider the input sequence $\mathbf{X} := [\mathbf{x}_1, \mathbf{x}_2, \ldots, \mathbf{x}_n]^\top \in \mathbb{R}^{n \times d}$, where $n$ is the sequence length and $d$ is the data dimension. The Transformer utilizes a multi-head attention mechanism to process this sequential input, allowing the model to learn correlations between different parts of the input sequence using trainable representations.

(i) In the multi-head attention framework, the input sequence $\mathbf{X}$ is first, for example, linearly transformed into $h$ different sets of keys, queries, and values, corresponding to $h$ attention heads. Specifically, for each head $i \in [h]$, we have $\mathbf{K}^{(i)} = \mathbf{X}\mathbf{W}_k^{(i)}, \mathbf{Q}^{(i)} = \mathbf{X}\mathbf{W}_q^{(i)}, \mathbf{V}^{(i)} = \mathbf{X}\mathbf{W}_v^{(i)} \in \mathbb{R}^{n \times d_h}$, where $\mathbf{W}_k^{(i)}, \mathbf{W}_q^{(i)}, \mathbf{W}_v^{(i)} \in \mathbb{R}^{d \times d_h}$ are trainable weight matrices for each head. Here, $d_h$ is the head dimension, and it typically holds that $d = d_h \times h$.

(ii) Then, for each head $i \in [h]$, the self-attention score and subsequent output are computed as
$$\mathbf{Attn}^{(i)}(\mathbf{X}) := \mathrm{softmax}\left(\frac{\mathbf{Q}^{(i)}\mathbf{K}^{(i)^\top}}{T}\right) \in \mathbb{R}^{n \times n}, \mathbf{Output}^{(i)} = \mathbf{Attn}^{(i)}(\mathbf{X})\mathbf{V}^{(i)}.$$

(iii) Next, all heads' outputs are concatenated and linearly transformed to yield the output of one multi-head attention layer, i.e. $\mathbf{MultiHeadAttn}(\mathbf{X}) = \mathrm{Concat}(\mathbf{Output}^{(1)}, \cdots, \mathbf{Output}^{(h)})\mathbf{W}_o$, where $\mathbf{W}_o \in \mathbb{R}^{hd_h \times d}$ is another trainable weight matrix.

(iv) Finally, the above output $\mathbf{MultiHeadAttn}(\mathbf{X})$ is passed through subsequent layers, including e.g. normalization layers and feed-forward neural networks, to produce the final output of the Transformer model.

## 3 FUNDAMENTAL SIMULATIONS

In this section, we provide detailed experiments on the most general Transformers in various settings to examine the rank of attention matrices. To facilitate comparisons and analysis, we report the ratio of attention ranks over sequence lengths (rank/seq len) rather than the absolute rank values to eliminate the interference caused by varied sizes of attention matrices across different sequence lengths.

### 3.1 BASIC PHENOMENA

First, we test for the most general Transformer models to examine the attention ranks under various head dimensions.

**Model** We use a standard one-layer Transformer encoder block with $d_{\mathrm{model}} = d = 384$ and a feed-forward hidden dimension of 512. We select the head dimension $d_h \in \{2, 4, 8, 16, 32, 64, 96\}$. The trainable weights are i.i.d. initialized using a standard normal distribution $\mathcal{N}(0, 1)$.

**Data** We generate random matrices with i.i.d. entries following the standard normal distribution $\mathcal{N}(0,1)$ with a shape of $(n,b,d)$, where the sequence length $n$ is set as 100, the batch size $b$ is 32 and the data dimension $d$ is 384. Subsequently, we record the mean and standard deviation of all $hb$ attention matrices for every $d_h$.

**Rank calculation** There are several equivalent definitions of the matrix rank in algebra. For numerical computation, the rank is usually calculated via the singular value decomposition (SVD), i.e., the rank equals to the number of non-zero singular values. In practice, due to the numerical precision limitation and round-off errors, this procedure often requires a relaxation, where a tolerance threshold $\epsilon$ is applied to yield the so-called numerical matrix rank. That is, $\text{rank}(\mathbf{A},\epsilon)$ equals to the number of singular values no less than $\epsilon$. Here, we set the tolerance threshold as $\epsilon = 10^{-8}$.

Table 1: Fundamental experimental results. The column labeled $d_h$ contains different head dimensions. The "Rank / Seq Len" represents the ratio of attention ranks over sequence lengths, with the standard deviation denoted by $\pm$. The "Improvement" column summarizes the successive increases in the "Rank / Seq Len" column compared to the previous row.

| $d_h$ | Rank / Seq Len | Improvement |
|---|---|---|
| 2 | $0.115 \pm 0.024$ | - |
| 4 | $0.255 \pm 0.032$ | + 0.140 |
| 8 | $0.404 \pm 0.035$ | + 0.149 |
| 16 | $0.508 \pm 0.039$ | + 0.104 |
| 32 | $0.569 \pm 0.033$ | + 0.061 |
| 64 | $0.603 \pm 0.031$ | + 0.034 |
| 96 | $0.622 \pm 0.034$ | + 0.019 |
| 192 | $0.632 \pm 0.028$ | + 0.010 |

**Observations** The experimental results summarized in Table 1 illustrate a clear relationship between the head dimension $d_h$ and Rank / Seq Len.

(i) For relatively small values of $d_h$, the attention matrix exhibits a low rank. As $d_h$ increases, significant increments of ranks are observed: when $d_h = 2$, Rank / Seq Len is around 0.11. When $d_h$ increases to 4, there is a notable increase in Rank / Seq Len to around 0.25.

(ii) For appropriately large values of $d_h$, further increases in $d_h$ lead to diminishing increments of attention ranks, with a final barrier of approximately $0.63n \ll n$ ($n$: the full-rank).

(iii) Although Rank / Seq Len increases with the head dimension $d_h$, the rate of this increment gradually decreases. For instance, Rank / Seq Len increases from around 0.40 at $d_h = 8$ to around 0.51 at $d_h = 16$, with an increment of 0.11. However, as $d_h$ further rises to 32, 64 and 96, the increments in Rank / Seq Len reduce to 0.06, 0.03 and 0.01, respectively. This suggests a more significant plateauing effect at higher $d_h$ levels.

(iv) The variances in Rank / Seq Len exhibit slight fluctuations across different $d_h$ values but remain relatively low, showing the stability of our experimental results.

The observations are summarized as follows.

- The attention rank increases with the head dimension $d_h$. When $d_h$ increases within relatively small values, there is a notable rise in the attention rank.

- When $d_h$ is appropriately large, further increases in $d_h$ result in only marginal increments of attention ranks, which is capped at around $0.63n \ll n$ (the full-rank).

### 3.2 ABLATION STUDIES ON MODELS

**Model dimensions** We start by investigating the effect of different model dimensions $d_{\text{model}} \in \{384, 768, 1152, 1536\}$, maintaining other configurations specified in Section 3.1. The results (provided in Appendix C.1) align with the phenomena observed in Figure 1 and Table 1, indicating a robust and consistent pattern of attention ranks across varied model dimensions.

**Softmax temperatures**   We test for the softmax temperature $T \in \{10^{-5}, 10^{-3}, 10^{-1}, 1\}$ to assess its effect on the attention rank. Similarly, the outcomes (detailed in Appendix C.2) also exhibit a robust and consistent pattern of attention ranks across different softmax temperatures.

**Transformers' layers**   To study the attention ranks in different layers, we test for a 8-layer Transformer. The results (elaborated in Appendix C.3) reveal a consistent pattern across different layers, with deeper layers appearing more pronounced low ranks.

### 3.3   ABLATION STUDIES ON DATASETS

**Sequence lengths**   We examine the influence of sequence lengths on attention ranks by varying the sequence lengths in $\{25, 50, 100, 200\}$. To ensure a comprehensive investigation, we employ a refined partition over the head dimension ($d_h \in \{2, 4, 8, 16, 32, 48, 64, 80, 96\}$) and increase the model dimension to $d_{\text{model}} = 960$. The other configurations remain the same as those outlined in Section 3.1. The results summarized in Table 2 imply a consistent pattern of attention ranks across various sequence lengths, confirming the robustness of our findings in Section 3.1 and Section 3.2. Notably, as is highlighted in Table 2, the required head dimensions for the saturation of attention ranks exhibit a linear increase with doubling sequence lengths, suggesting a potential logarithmic dependency.

**Data distributions**   We also investigate attention ranks under different types of data distributions, including $\mathcal{N}(0, 1)$, $\mathcal{N}(0, 100)$, $\mathcal{U}(-1, 1)$ and $\mathcal{U}(-100, 100)$, and consistent phenomena irrespective of data distributions are observed. For comprehensive discussions and detailed experimental reports, refer to Appendix C.4. These results, aligning with those in previous sections, underscore the robustness of our findings with respect to data distributions.

Table 2: The attention ranks for different sequence lengths. Here, $d_h$ represents the head dimension. The highlighted boldface statistics are selected according to the "Improvement" column: when the improvement drops less than or around 0.01 for the first time at a certain row, we select the *above* one row as the critical position of $d_h$ where the saturation of attention ranks begins to occur. One can observe that as the sequence length doubles, the required head dimension to reach the saturation increases linearly, which potentially implies certain $\log$-dependence.

|  | Seq Len = 25 | | Seq Len = 50 | | Seq Len = 100 | | Seq Len = 200 | |
|---|---|---|---|---|---|---|---|---|
| $d_h$ | Rank/Seq Len | Improvement | Rank/Seq Len | Improvement | Rank/Seq Len | Improvement | Rank/Seq Len | Improvement |
| 2 | $0.250 \pm 0.051$ | - | $0.158 \pm 0.029$ | - | $0.096 \pm 0.019$ | - | $0.055 \pm 0.011$ | - |
| 4 | $0.422 \pm 0.061$ | +0.172 | $0.324 \pm 0.044$ | +0.166 | $0.240 \pm 0.032$ | +0.144 | $0.172 \pm 0.019$ | +0.117 |
| 8 | $0.530 \pm 0.068$ | +0.108 | $0.459 \pm 0.047$ | +0.135 | $0.391 \pm 0.035$ | +0.151 | $0.323 \pm 0.025$ | +0.151 |
| 16 | $\mathbf{0.606 \pm 0.055}$ | +0.076 | $0.536 \pm 0.052$ | +0.077 | $0.498 \pm 0.029$ | +0.107 | $0.443 \pm 0.026$ | +0.120 |
| 32 | $0.612 \pm 0.066$ | +0.006 | $\mathbf{0.593 \pm 0.045}$ | +0.057 | $0.571 \pm 0.031$ | +0.073 | $0.525 \pm 0.023$ | +0.082 |
| 48 | $0.618 \pm 0.048$ | +0.006 | $0.601 \pm 0.033$ | +0.008 | $\mathbf{0.594 \pm 0.034}$ | +0.023 | $0.554 \pm 0.018$ | +0.029 |
| 64 | $0.621 \pm 0.060$ | +0.003 | $0.612 \pm 0.057$ | +0.011 | $0.606 \pm 0.038$ | +0.012 | $\mathbf{0.579 \pm 0.021}$ | +0.025 |
| 80 | $0.623 \pm 0.071$ | +0.002 | $0.615 \pm 0.054$ | +0.003 | $0.609 \pm 0.049$ | +0.003 | $0.592 \pm 0.018$ | +0.013 |
| 96 | $0.625 \pm 0.058$ | +0.002 | $0.622 \pm 0.058$ | +0.007 | $0.611 \pm 0.034$ | +0.002 | $0.597 \pm 0.020$ | +0.005 |

For more general cases, such as real-world datasets, more types of distributions and non-i.i.d. data, one can check Figure 2 for details. It is observed that the above phenomena still hold in general.

## 4   THEORETICAL ANALYSIS

In this section, we provide the fine-grained mathematical analysis to demonstrate rigorously the experimental results reported in Section 3, i.e. the existence of the low-rank barrier and model-reduction effect.

### 4.1   PRELIMINARIES

For clarity, we restate the requisite notations here. Recall that $\mathbf{X} = [\mathbf{x}_1, \mathbf{x}_2, \ldots, \mathbf{x}_n]^\top \in \mathbb{R}^{n \times d}$ is the input sequence, where $n$ denotes the sequence length and $d$ is the input dimension. Without loss of generality, we focus on one head. Let $(\mathbf{K}, \mathbf{Q}) = (\mathbf{X}\mathbf{W}_k, \mathbf{X}\mathbf{W}_q)$ be the key-query pair with trainable parameters $\boldsymbol{\theta} := (\mathbf{W}_k, \mathbf{W}_q) \in \mathbb{R}^{d \times d_h} \times \mathbb{R}^{d \times d_h}$ ($d_h$ is the head dimension), i.e., $\mathbf{K} :=$

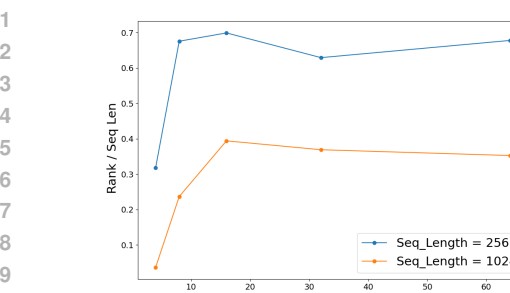 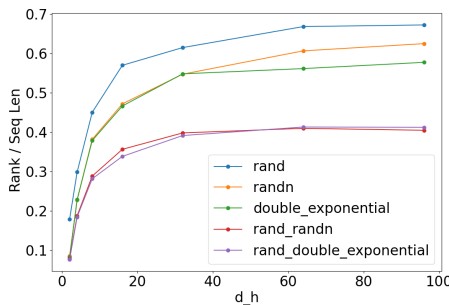

Figure 2: Left: We conduct experiments on the CIFAR-10 dataset to verify the effect of sequence lengths. By adjusting the patch size, we can accordingly change the input sequence length. It is observed that even with extended sequence lengths (from 256 to 1024), analogous patterns remain evident. Right: Similar patterns hold for more distributions and non-i.i.d. data. The `rand_randn` line represents tensors where half of the elements are sampled from a uniform distribution and the other half from a Gaussian distribution. The `rand_double_exponential` line denotes tensors where half of the elements are sampled from a uniform distribution and the other half from a double exponential distribution.

$[\mathbf{k}_1, \mathbf{k}_2, \ldots, \mathbf{k}_n]^\top \in \mathbb{R}^{n \times d_h}$, $\mathbf{Q} := [\mathbf{q}_1, \mathbf{q}_2, \ldots, \mathbf{q}_n]^\top \in \mathbb{R}^{n \times d_h}$ with $\mathbf{k}_i^\top = \mathbf{x}_i^\top \mathbf{W}_k$, $\mathbf{q}_i^\top = \mathbf{x}_i^\top \mathbf{W}_q$, $i = 1, 2, \ldots, n$. The basic form of the self-attention score matrix is defined as

$$\mathbf{Attn}(\mathbf{X}; \boldsymbol{\theta}) := \mathrm{softmax}\left(\mathbf{Q}\mathbf{K}^\top / T\right) = \mathrm{softmax}\left(\mathbf{X}\mathbf{W}_q \mathbf{W}_k^\top \mathbf{X}^\top / T\right), \tag{1}$$

where $T > 0$ is the temperature. By convention, for any $\mathbf{A} = [a_{ij}] \in \mathbb{R}^{n \times n}$, $\mathbf{e}_i^\top \mathrm{softmax}(\mathbf{A})\mathbf{e}_j := \frac{\exp(a_{ij})}{\sum_{j=1}^n \exp(a_{ij})}$ with $\{\mathbf{e}_i\}_{i=1}^n$ as the standard basis of $\mathbb{R}^n$.

Since $\mathbf{K}, \mathbf{Q} \in \mathbb{R}^{n \times d_h}$, we get $\mathbf{Q}\mathbf{K}^\top / T \in \mathbb{R}^{n \times n}$, and hence the trivial upper bound $\mathrm{rank}\left(\mathbf{Attn}(\mathbf{X}; \boldsymbol{\theta})\right) \leq n$. We further deduce that

$$\mathrm{rank}\left(\mathbf{Q}\mathbf{K}^\top / T\right) = \mathrm{rank}\left(\mathbf{Q}\mathbf{K}^\top\right) \leq \min\{\mathrm{rank}(\mathbf{Q}), \mathrm{rank}(\mathbf{K}^\top)\} \leq \min\{n, d_h\} = d_h, \tag{2}$$

with the typical configuration $n > d_h$ in practice. Intuitively, one may expect that for any (or most) $d_h$, $\mathrm{rank}\left(\mathrm{softmax}\left(\mathbf{Q}\mathbf{K}^\top / T\right)\right) \gg d_h$, even $\mathrm{rank}\left(\mathrm{softmax}\left(\mathbf{Q}\mathbf{K}^\top / T\right)\right) \approx n$ due to the injection of nonlinearity. The experimental results in Section 3 also support this intuition for relatively small $d_h$. However, this is not the case when $d_h$ is appropriately large. In the following section, we provide theoretical results to rigorously analyze these phenomena.

## 4.2 Main Results

In this section, we give a fine-grained theoretical characterization of the low-rank barrier and model-reduction effect. That is, (i) there exists a non-trivial upper bound ($\approx 0.63n$) of the attention rank (i.e. $\mathrm{rank}\left(\mathbf{Attn}(\mathbf{X}; \boldsymbol{\theta})\right)$) in expectation regardless of the head dimension $d_h$; (ii) $\mathrm{rank}\left(\mathbf{Attn}(\mathbf{X}; \boldsymbol{\theta})\right)$ gets saturation when $d_h = \Omega(\log n)$.

For convenience, we focus on the low-temperature case ($T > 0$ appropriately small) associated with the "hardmax" activation. Note that although this setup is established for theoretical simplicity, the hardmax activation is occasionally used in applications for computational efficiency. See computer vision (CV) examples in Elsayed et al. (2019); Papadopoulos et al. (2021) for more details.

For the low-temperature case with $T > 0$ appropriately small, the right hand side of (1) is approximately

$$\mathrm{hardmax}\left(\mathbf{X}\mathbf{W}_q \mathbf{W}_k^\top \mathbf{X}^\top\right), \tag{3}$$

where the maximum is also taken in a row-wise sense: for a matrix $\mathbf{A} = [a_{ij}] \in \mathbb{R}^{n \times n}$, $\mathbf{e}_i^\top \mathrm{hardmax}(\mathbf{A}) := \mathbf{e}_{k_i}$ with $k_i := \arg\max_{j \in [n]} a_{ij}$. Note that the $\mathrm{hardmax}(\cdot)$ operator is positively scaling-invariant, i.e. $\mathrm{hardmax}(c\mathbf{A}) = \mathrm{hardmax}(\mathbf{A})$ for any $c > 0$.

**Remark 1.** *Numerically, we have demonstrated in Figure 6 that the attention rank of Transformers is robust to variations in softmax temperatures, as least in the range between low temperatures (hardmax) and normal temperatures (softmax). In this work, all the experiments are performed for normal temperatures, obtaining consistent results with the following theories.*

We have the following main theorem to estimate the (averaged) rank of (3). The derived upper bound (proofs deferred in Appendix B) coincides perfectly with the experimental results (see details in Figure 1 and Table 1).

**Theorem 1.** *Let the parameters $\mathbf{W}_q, \mathbf{W}_k$ be Gaussian random matrices, i.e., the entries of $\mathbf{W}_q, \mathbf{W}_k$ are independent $\mathcal{N}(0,1)$ random variables. Assume that the input sequence $\mathbf{X}$ satisfies $\mathbf{X}\mathbf{X}^\top = \mathbf{I}_n$. Then for any $n \in \mathbb{N}_+$ appropriately large, we have*

$$\mathbb{E}_{\mathbf{W}_k, \mathbf{W}_q} \left[ \text{rank} \left( \text{hardmax} \left( \mathbf{X}\mathbf{W}_q\mathbf{W}_k^\top\mathbf{X}^\top \right) \right) \right] \leq (1 - \exp(-1))n + O(1) \approx 0.63n. \quad (4)$$

**Remark 2.** *Theoretically, the (exact) orthonormality assumption of input sequences in Theorem 1 can be relaxed to the almost orthonormality via approximation procedures and stability/perturbation analysis. See details in Section B.1.*

**Remark 3.** *The assumption that the input sequence is (almost) orthonormal might seem stringent at the first glance. However, in practical scenarios, particularly in high-dimensional spaces ($d \gg 1$), the (embedding) vectors (denoted as $\mathbf{x}_i$) representing different tokens are often almost orthogonal, since they are typically modeled using independent, isotropic Gaussian random vectors.[1] This assumption is also proposed by Tian et al. (2024) (a theoretical paper to analyze the training dynamics of Transformers). According to Tian et al. (2024), the almost orthogonality even holds during the training process (for large pre-trained models such as Pythia, BERT, OPT, LLaMA and ViT of different sizes, see details in Tian et al. (2024), Appendix B.1). We also numerically verify the orthonormality by ourselves in Appendix C.5 (Figure 8) on both synthetic and real-world datasets.*

**Remark 4.** *Recall that the* hardmax *operator is invariant under the positive scaling. Consequently, Theorem 1 remains valid even in cases where input sequences are not normalized. This property underscores the robustness of the* hardmax *operation in various input conditions.*

**The low-rank bottleneck on approximation** According to Eckart–Young theorem (Eckart & Young, 1936), there exists a lower bound corresponding to the spectral regularity of approximated (target) matrices for the low-rank approximation problem. For instance, given the target matrix $\mathbf{A} \in \mathbb{R}^{n \times n}$ with singular values $\sigma_1 \geq \cdots \geq \sigma_{n'} > \sigma_{n'+1} = \cdots = \sigma_n = 0$ (i.e. $\text{rank}(\mathbf{A}) = n' \in [0.64n, n]$), based on Eckart–Young theorem and Theorem 1, we have $\left\| \text{hardmax} \left( \mathbf{Q}\mathbf{K}^\top \right) - \mathbf{A} \right\|_F^2 \geq \sum_{i=\text{rank}(\text{hardmax}(\mathbf{Q}\mathbf{K}^\top))+1}^{n'} \sigma_i^2 \overset{\text{e}}{\geq} \sum_{i=(1-\exp(-1))n+O(1)}^{n'} \sigma_i^2 \approx$

$\sum_{i=0.63n}^{n'} \sigma_i^2 > 0$ for any $n \in \mathbb{N}_+$ appropriately large, where $\overset{\text{e}}{\geq}$ represents "no less than" in expectation. One can expect that this lower bound implies a large gap if $\{\sigma_i\}_{i=1}^n$ (the spectrum of $\mathbf{A}$) decays slowly (e.g. $\mathbf{A}$ has a full rank $n$).

**The model-reduction effect** In fact, the above rank (the left hand side of (4)) reaches saturation when continuously increasing the head dimension $d_h$, provided an appropriate scaling (e.g. $1/\sqrt{d_h}$). Recall that the rows of $\mathbf{X}\mathbf{W}_q\mathbf{W}_k^\top\mathbf{X}^\top = \mathbf{Q}\mathbf{K}^\top$ are independent and identically distributed as $\mathcal{N}(\mathbf{0}_n, \mathbf{K}\mathbf{K}^\top)$, according to Johnson–Lindenstrauss lemma (Johnson & Lindenstrauss, 1984), we have

$$\mathbf{e}_i^\top \mathbf{K}\mathbf{K}^\top\mathbf{e}_j = \mathbf{k}_i^\top\mathbf{k}_j = \mathbf{x}_i^\top\mathbf{W}_k\mathbf{W}_k^\top\mathbf{x}_j \approx d_h\mathbf{x}_i^\top\mathbf{x}_j \quad (5)$$

with high probabilities when $d_h = \Omega(\log n)$, which gives

$$\mathbf{e}_i^\top\mathbf{Q}\mathbf{K}^\top/\sqrt{d_h} \sim \mathcal{N}(\mathbf{0}_n, \mathbf{K}\mathbf{K}^\top/d_h) \approx \mathcal{N}(\mathbf{0}_n, \mathbf{X}\mathbf{X}^\top), \quad d_h = \Omega(\log n). \quad (6)$$

---

[1]As is shown in Vershynin (2018) (specifically, Lemma 3.2.4 and Remark 3.2.5), these vectors exhibit near-orthogonality after an appropriate scaling such as normalization.

Due to the (positive) scaling-invariant property of hardmax, we approximately deduce that the above rank (the left hand side of (4)) only depends on $\mathbf{X}$ (and hence $n, d$), i.e.

$$\text{rank}\left(\text{hardmax}\left(\mathbf{X}\mathbf{W}_q\mathbf{W}_k^\top\mathbf{X}^\top\right)\right) = \text{rank}\left(\text{hardmax}\left(\mathbf{Q}\mathbf{K}^\top/\sqrt{d_h}\right)\right) \tag{7}$$

$$\overset{\text{d}}{\approx} \text{rank}\left(\text{hardmax}\left(\text{rows of } \mathcal{N}(\mathbf{0}_n, \mathbf{X}\mathbf{X}^\top)\right)\right), \tag{8}$$

when $d_h = \Omega(\log n)$, where $\overset{\text{d}}{\approx}$ represents the approximation in distribution. That is, increasing the head dimension beyond a certain threshold, specifically after $d_h^* = \Omega(\log n)$, results in a *limited* impact on the attention rank, which is eventually influenced by $n$ and $d$. This phenomenon can be understood as a manifestation of the model-reduction effect: selecting the critical configuration $d_h^* = \Omega(\log n)$ achieves optimal model efficiency, since further increasing parameters leads to *diminishing marginal utility*.

**Remark 5.** *For the constants involved in $d_h = \Omega(\log n)$, according to Johnson–Lindenstrauss lemma, it is of order $1/\epsilon^2$, where $\epsilon$ is the gap tolerance between the products of projected vectors and original vectors (i.e. the error of "$\approx$" in (5)). Additionally, there are universal constants related to $\delta$ (probability tolerance) and methods of projections. That is, for requirements of higher probabilities (smaller $\delta$), the universal constants are larger; for nonlinear projections instead of linear random projections used here, the universal constants can be potentially smaller.*

### 4.3 DISCUSSIONS

In this section, we revisit the experimental results in Section 3, and compare them with the developed theoretical results in Section 4.2. Comparing the estimates (4) and (8) (with $d_h = \Omega(\log n)$) with the observations in Section 3, we obtain the *consistency* between our theoretical results and simulation outcomes.

First, considering Figure 1, 5, 6, 7 and Table 1, 2, 3, we note that under various settings (such as the model dimension, softmax temperature, model depth, sequence length and data distribution), the attention rank increases with the head dimension $d_h$, yet it converges towards the upper bound predicted by the estimates (4). Furthermore, the incremental growth of the attention rank significantly diminishes with a uniform increase in $d_h$, indicating an obvious trend towards the saturation.

Second, we focus on Table 2, which not only facilitates a detailed analysis of the rank saturation point, but also quantitatively corroborates the estimate (8) with $d_h = \Omega(\log n)$. Based on the highlighted boldface statistics, it is evident that for *doubled* sequence lengths, a distinct *linear increment* trend of head dimensions is observed in the saturation positions. For instance, at the sequence length of 25, the saturation occurs at $d_h = 16$. As a comparison, for sequence lengths of 50, 100 and 200, the critical positions of saturation are identified at $d_h = 32, 48$ and $64$, respectively. This finding aligns with the theoretical estimate (8) with $d_h = \Omega(\log n)$: the critical saturation position ($d_h^*$) exhibits a linear escalation corresponding to the exponential increase in the sequence length $n$.

## 5 REAL-WORLD EXPERIMENTS

In this section, we further verify our previous findings through simulations on real-world datasets. In theory, the upper bound is derived for every single head (with randomly initialized parameters). For the multiple heads case, we aim to emphasize the *saturation* effect via numerical simulations. That is, despite that one can increase the overall rank by concatenation, the low-rank saturation of every single head still leads to an *inefficiency* issue: As is shown later, both the attention rank and model performance *consistently* get *marginal enhancements* when increasing parameters, implying the model redundancy. This gives chances for the optimal configuration of hyper-parameters: In practical applications, one may check the saturation situation of attention ranks before training, and set the optimal number of parameters as where the rank first gets saturated.

### 5.1 LOW-RANK BARRIER VERIFICATIONS

**Setup** The experiments focus on evaluating the performance of Vision Transformers (ViTs; Dosovitskiy et al. (2021)) on image classification tasks, e.g. using the CIFAR-100 dataset. We perform

the train-validation-test split on the datasets following official guidelines. Here, we set the model dimension $d_{\text{model}} = 384$, and also the feed-forward hidden dimension as $384$. The model depth is 7. For the learning, the batch sizes are $128$ for training and $1024$ for evaluation. The initial learning rate is set as $10^{-3}$. To align with real-world applications, various techniques are integrated, including label smoothing and auto-augmentation. Moreover, the experiments also involve advanced regularization methods (specifically, CutMix (Yun et al., 2019) and MixUp (Zhang et al., 2018) to enhance the models' generalization performance.

**Analysis** In this series of experiments, we fix the input/model dimension $d = d_{\text{model}}$, and vary the number of heads $h$, the head dimension $d_h$ following the equation $d = h \times d_h$, which is default in practical applications. With this constraint, a smaller number of heads $h$ results in a larger head dimension $d_h$, potentially exceeding the necessary head dimensions to get the rank saturation for each head. Equivalently, most of heads may have reached the saturation point, leading to the parameter redundancy. However, as the number of heads increases, the Transformer model with reduced head dimensions gradually avoids the rank saturation (and potential parameter redundancy), leading to more portions of "effective" ranks for modeling, which yields improved experimental results. Figure 3a shows that increasing the number of heads ($h = 1, 2, 4, 8$) benefits the model's performance in general, and the corresponding attention ranks in Figure 3b are already saturated (for $d_h = 384, 192, 96, 48$), aligning with the above arguments.

In addition, there are also analogous observations on the CIFAR-10 and SVHN dataset under different input/model dimensions (see more details in Appendix D.1).

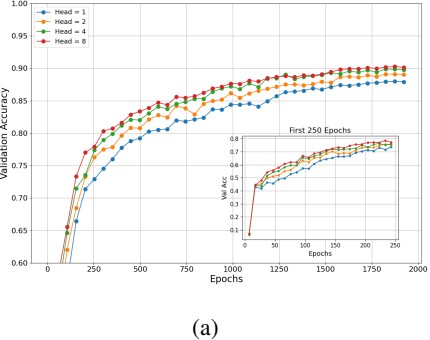 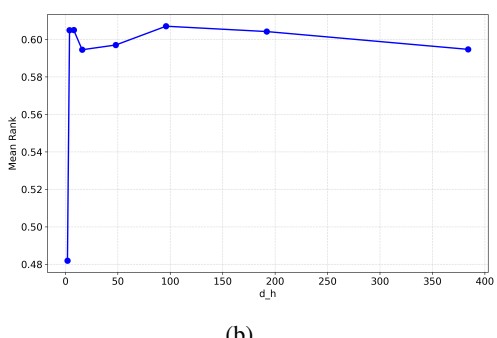

(a)                                                 (b)

Figure 3: Real-world experiments on the CIFAR-10 dataset. (a): The validation accuracy across training epochs for different numbers of heads ($h$) with a fixed input/model dimension ($d = d_{\text{model}} = 384$). The inset provides a magnified view of the first 250 epochs to emphasize the early training dynamics. (b): The corresponding attention ranks, which are calculated for the first-layer attention matrices on a mini-batch of CIFAR-10 images (averaged over both all heads and multiple repeated random seeds) under different numbers of heads. For $h = 1, 2, 4, 8$, the corresponding $d_h = 384, 192, 96, 48$. It is observed that under these configurations, the mean attention matrix ranks are saturated, hence decreasing $d_h$ will not affect the expressive ability of each head, and the model performance will instead improve from an increase in the number of heads.

## 5.2 MODEL-REDUCTION EFFECT VERIFICATIONS

In this section, the primary setup of experiments is the same as that of Section 5.1. This allows us to scrutinize the effect of reducing the model's dimension on performance metrics. Figure 4a illustrates the experiments conducted on the CIFAR-10 dataset, particularly with the number of heads $h = 8$. It is shown that although the initial improvement in the validation accuracy is pronounced as the head dimension $d_h$ increases within relatively small values, this improvement plateaus for appropriately large values of $d_h$, showcasing diminishing returns with further increments in model parameters. This observation corroborates our theoretical justifications on the model-reduction effect, suggesting an optimal range of head dimensions that balance the model performance with parameter efficiency. In Figure 4a, the optimal $d_h^* = 16$, since $d_h = 32$ yields marginal improvements in accuracies.

Notably, the corresponding attention ranks in Figure 4b *also* appear the saturation when $d_h \geq d_h^* = 16$, which *aligns* with the marginal performance improvements (i.e. $d_h = 16, 32$ in Figure 4a).

Additionally, there are also similar results on the CIFAR-100, SVHN and IMDB dataset under various head dimensions and different input sizes. See more detailed experimental outcomes in Appendix D.2 and Appendix D.3.

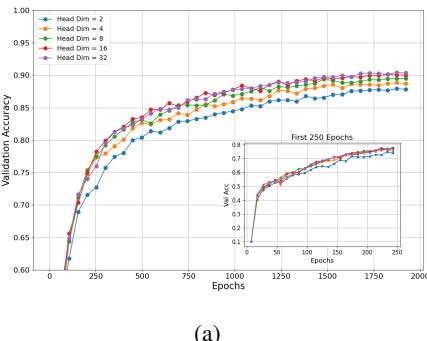
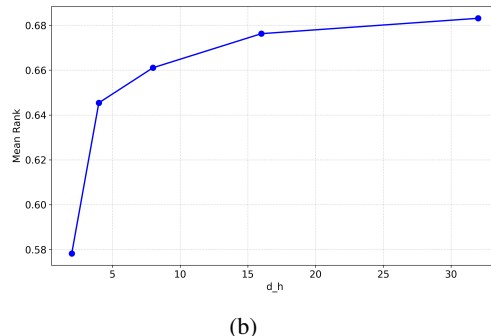

(a)                  (b)

Figure 4: Real-world experiments on the CIFAR-10 dataset. (a): The validation accuracy across training epochs for different head dimensions ($d_h$) with a fixed number of heads ($h = 8$). The inset provides a magnified view of the first 250 epochs to emphasize the early training dynamics. (b): The corresponding attention ranks, which are calculated for the first-layer attention matrices on a mini-batch of CIFAR-10 images (averaged over both all heads and multiple repeated random seeds) under different head dimensions (and hence different model dimensions). We test for 5 different values of $d_h$: $d_h = 2, 4, 8, 16, 32$. We observe a similar pattern with Figure 1, where smaller values of $d_h$ lead to significant improvements in attention ranks as $d_h$ increases. However, when the values of $d_h$ become larger ($d_h \geq 16$), its further increases have marginal effects on attention ranks. Additionally, the variation trend of attention ranks *aligns* with that of model performance in Figure 4a. That is, although an increase in attention ranks positively correlates with improved model performance, both of the ranks and performance get saturated *simultaneously* (i.e. at $d_h^* = 16$), implying the optimal parameter efficiency around $d_h^*$.

## 6 CONCLUSION

In this research, we present an extensive investigation into the rank of the attention matrix in Transformer architectures, drawing insights from both theoretical analysis and empirical observations. From a theoretical perspective, we derive a clear upper bound on the attention rank, approximately $\approx 0.63n$, which is notably lower than the full rank $n$, revealing the existence of a low-rank constraint. Furthermore, we quantitatively show that for relatively small head dimensions $d_h = \Omega(\log n)$, the attention rank approaches saturation, implying that further increases in model parameters provide diminishing returns in performance (model-reduction effect). From an experimental perspective, we validate these theoretical insights by conducting a comprehensive set of tests involving various model architectures and diverse real-world datasets. These experiments confirm the validity and robustness of our theoretical insights, demonstrating their applicability to a wide array of scenarios. This developed relationship between head dimensions and attention ranks provides deeper understandings and valuable insights into the evaluation of general Transformer models' capacity and efficiency.

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

## A  RELATED WORK

The exploration of the rank of the Transformer attention matrix has been a focus in previous research (Kanai et al., 2018; Bhojanapalli et al., 2020; Dong et al., 2021; Lin et al., 2022). Bhojanapalli et al. (2020) unveiled a restriction associated with the low-rank bottleneck in attention heads, attributed to the proportional relationship between the number of heads and the size of each head in prevailing architectures. Dong et al. (2021) introduced an innovative perspective of interpreting self-attention networks. Their study elucidated that the networks' output is an amalgamation of lesser components, or pathways. In the absence of skip connections and multi-layer perceptrons (MLPs), they established that the output gravitates towards a rank-1 matrix at a doubly exponential rate.

On the other hand, a suite of Transformer-based adaptations (Chen et al., 2021a; Wang et al., 2020; Hu et al., 2022; Guo et al., 2019; Lin et al., 2022) has emerged to mitigate the inherent bottlenecks, notably computational and memory constraints. For instance, Wang et al. (2020) ascertained that the self-attention mechanism's complexity is reducible, attributing this to its low-rank matrix approximation. The innovative self-attention mechanism they introduced marked a reduction in complexity. Meanwhile, Guo et al. (2019) incorporated low-rank constraints, a modification that manifested improvements in specific tasks. In a parallel vein, Chen et al. (2021a) noted the prowess of sparse and low-rank approximations in distinct scenarios. Their efficacy was found to be contingent on the softmax temperature in attention, with a combined sparse and low-rank approach superseding individual performances. Another line of work focuses on the computation efficiency of Transformer models, e.g. KDEformer (Zandieh et al., 2023) and HyperAttention (Han et al., 2024). These works studied the approximate calculation problem of attention matrices (with direct applications in model compression), with the fundamental approach to reduce the full matrix multiplication to sub-matrix multiplications, and relate to attention ranks through the size of sub-matrices, which is typically lower bounded by measures depending on (stable) ranks of attention matrices. It would be interesting to further develop these works with the inductive biases established here, i.e. explore potentially more efficient algorithms given the low-rank barrier and rank saturation of attention matrices.

As a comparison, this study explores the ranks of attention score matrices in Transformers, and reveals two main insights: although the attention rank grows with the head dimension, it has an upper limit (*low-rank barrier*). Additionally, a *model-reduction effect* is uncovered. These phenomena are consistent across different configurations for both models and (real-world) datasets, and also rigorously proved with aligned theoretical characterizations.

## B  PROOFS

In this section, we provide all the missing proofs. The proof entails a detailed analysis of matrix operations, probability transforms, and infinitesimal order estimation. Specifically, the proof proceeds as follows:

1. Given the orthonormal nature of input sequences, according to Lemma 4, one can derive that different rows of $\mathbf{X}\mathbf{W}_q\mathbf{W}_k^\top\mathbf{X}^\top$ are independent, and these rows are identically distributed as $\mathcal{N}(\mathbf{0}_n, \mathbf{K}\mathbf{K}^\top)$, conditioned on any fixed Gaussian random matrix $\mathbf{W}_k$.

2. Note that applying the hardmax operation to individual rows is analogous to solving an elementary birthday problem (refer to Lemma 3), which reduces the original problem as counting columns with all zeros.

3. The estimate is further refined based on Lemma 2, and completed by applying the AM-GM inequality, which indicates the equality when all probabilities are equal.

To begin with, the key approximation (3) is due to the following lemma, which characterizes the gap between the softmax function and its "hard" version.

**Lemma 1.** *Let* $\mathbf{a} = [a_1, a_2, \cdots, a_n]^\top \in \mathbb{R}^n$ *with* $i^* := \arg\max_{i \in [n]} a_i$ *and* $i'^* := \arg\max_{i \in [n], i \neq i^*} a_i$, *and* $\text{hardmax}(\mathbf{a}) := \mathbf{e}_{i^*}$. *Assume that* $\delta := a_{i^*} - a_{i'^*} > 0$ *(i.e., the maximum is unique). Then for any* $T > 0$*, we have*

$$\Delta_{n,\delta}(T) := \|\text{softmax}(\mathbf{a}/T) - \text{hardmax}(\mathbf{a})\|_1$$
$$\leq 2(n-1)\exp(-\delta/T). \tag{9}$$

*That is, $\Delta_{n,\delta}(T)$ converges to 0 exponentially fast as $T \to 0^+$.*

*Proof.* It is straightforward to have

$$
\Delta_{n,\delta}(T) = \sum_{i \in [n], i \neq i^*} \frac{\exp(a_i/T)}{\sum_{j=1}^n \exp(a_j/T)} + 1 - \frac{\exp(a_{i^*}/T)}{\sum_{j=1}^n \exp(a_j/T)}
$$

$$
= 2 \frac{\sum_{i \in [n], i \neq i^*} \exp(a_i/T)}{\sum_{i \in [n], i \neq i^*} \exp(a_i/T) + \exp(a_{i^*}/T)}
$$

$$
\leq 2 \sum_{i \in [n], i \neq i^*} \exp((a_i - a_{i^*})/T)
$$

$$
\leq 2(n-1) \exp((a_{i'^*} - a_{i^*})/T)
$$

$$
= 2(n-1) \exp(-\delta/T). \tag{10}
$$

This gives $\lim_{T \to 0^+} \Delta_{n,\delta}(T) = 0$, and the rate is exponentially fast. The proof is completed. $\square$

Before we prove the low-rank barrier and model-reduction effect of (3), the following lemmas are useful.

**Lemma 2.** *For any $n \in \mathbb{N}_+$, define $\delta_n(p) := \exp(-pn) - (1-p)^n$, $p \in [0, +\infty)$. Then we have*

$$
\delta_n(p) \leq \frac{1}{2} p^2 n \exp(-p(n-1)) \tag{11}
$$

$$
\leq \begin{cases} \frac{1}{2} p^2, & n = 1, \\ 2 \exp(-2) \left( \frac{1}{n-1} + \frac{1}{(n-1)^2} \right), & n \geq 2. \end{cases} \tag{12}
$$

*Proof.* Note that $a_1^n - a_2^n = (a_1 - a_2) \sum_{k=0}^{n-1} a_1^{n-1-k} a_2^k$ for any $a_1, a_2 \in \mathbb{R}$, we have

$$
\delta_n(p) = (\exp(-p))^n - (1-p)^n = [\exp(-p) - (1-p)] \sum_{k=0}^{n-1} (\exp(-p))^{n-1-k} (1-p)^k. \tag{13}
$$

Let $g_1(p) := \exp(-p) - (1-p)$, $g_2(p) := \exp(-p) - (1-p+p^2/2) = g_1(p) - p^2/2$, $p \in [0, +\infty)$, we get

$$
g_1'(p) = -\exp(-p) + 1 \geq 0 \Rightarrow g_1(p) \geq g_1(0) = 0, \tag{14}
$$

$$
g_2'(p) = -\exp(-p) + 1 - p = -g_1(p) \leq 0 \Rightarrow g_2(p) \leq g_2(0) = 0, \tag{15}
$$

which gives

$$
\delta_1(p) = g_1(p) \leq p^2/2, \tag{16}
$$

$$
\delta_n(p) \leq \frac{1}{2} p^2 \sum_{k=0}^{n-1} (\exp(-p))^{n-1-k} (\exp(-p))^k = \frac{1}{2} p^2 n (\exp(-p))^{n-1}, \quad n \geq 2. \tag{17}
$$

For any $n \in \mathbb{N}_+$, $n \geq 2$, let $h_n(p) := p^2 (\exp(-p))^{n-1}$, $p \in [0, +\infty)$, we get $h_n'(p) = p(\exp(-p))^{n-1}(2 - p(n-1))$, hence

$$
h_n'(p) = 0 \Rightarrow p = 0 \text{ or } p = 2/(n-1) \Rightarrow h_n(p) \leq h_n(2/(n-1)) = \frac{4 \exp(-2)}{(n-1)^2}. \tag{18}
$$

Therefore

$$
\delta_n(p) \leq \frac{1}{2} n h_n(p) \leq \frac{2 \exp(-2)n}{(n-1)^2} = 2 \exp(-2) \left( \frac{1}{n-1} + \frac{1}{(n-1)^2} \right), \quad n \geq 2, \tag{19}
$$

which completes the proof. $\square$

**Lemma 3.** *For a random matrix $\mathbf{A} = [a_{ij}] \in \mathbb{R}^{n \times n}$ with independent rows, let $p_{ij} := \mathbb{P}(\{a_{ij} = \max_{j' \in [n]} a_{ij'}\})$. Then the expectation number of columns with all zeros in $\mathrm{hardmax}(\mathbf{A})$ is*

$$
\sum_{j=1}^n \prod_{i=1}^n (1 - p_{ij}). \tag{20}
$$

*Proof.* For $j = 1, 2, \ldots, n$, define the random variable

$$X_j = \begin{cases} 1, & \text{hardmax}(\mathbf{A})\mathbf{e}_j = \mathbf{0}_n, \\ 0, & \text{hardmax}(\mathbf{A})\mathbf{e}_j \neq \mathbf{0}_n. \end{cases} \tag{21}$$

By independence, we get

$$\mathbb{P}(\{X_j = 1\}) = \mathbb{P}\left( \bigcap_{i=1}^{n} \left\{ \mathbf{e}_i^\top \text{hardmax}(\mathbf{A})\mathbf{e}_j = 0 \right\} \right)$$

$$= \prod_{i=1}^{n} \mathbb{P}\left( \left\{ \mathbf{e}_i^\top \text{hardmax}(\mathbf{A})\mathbf{e}_j = 0 \right\} \right)$$

$$= \prod_{i=1}^{n} (1 - p_{ij}). \tag{22}$$

Therefore, the expectation number of columns with all zeros is

$$\mathbb{E}\left[ \sum_{j=1}^{n} X_j \right] = \sum_{j=1}^{n} \mathbb{E}[X_j] = \sum_{j=1}^{n} \mathbb{P}(\{X_j = 1\}) = \sum_{j=1}^{n} \prod_{i=1}^{n} (1 - p_{ij}), \tag{23}$$

which completes the proof. $\qquad\square$

The required independence in Lemma 3 is provided by the following lemma.

**Lemma 4.** *(Vershynin (2018), Exercise 3.3.6) Let $\mathbf{G} \in \mathbb{R}^{m \times n}$ be a Gaussian random matrix, i.e. the entries of $\mathbf{G}$ are independent $\mathcal{N}(0, 1)$ random variables. Let $\mathbf{u}, \mathbf{v} \in \mathbb{R}^n$ be unit orthogonal vectors. Then, $\mathbf{G}\mathbf{u}$ and $\mathbf{G}\mathbf{v}$ are independent $\mathcal{N}(\mathbf{0}_m, \mathbf{I}_m)$ random vectors.*

*Proof.* First, we show that $\mathbf{G}\mathbf{u}, \mathbf{G}\mathbf{v}$ are both $\mathcal{N}(\mathbf{0}_m, \mathbf{I}_m)$ random vectors. This is straightforward since $\mathbf{G}\mathbf{e}_j \sim \mathcal{N}(\mathbf{0}_m, \mathbf{I}_m)$ gives $u_j \mathbf{G}\mathbf{e}_j \sim \mathcal{N}(\mathbf{0}_m, u_j^2 \mathbf{I}_m)$, and $\{u_j \mathbf{G}\mathbf{e}_j\}_{j=1}^n$ is a collection of independent Gaussian vectors. Hence $\mathbf{G}\mathbf{u} = \sum_{j=1}^{n} u_j \mathbf{G}\mathbf{e}_j \sim \mathcal{N}(\mathbf{0}_m, \|\mathbf{u}\|_2^2 \mathbf{I}_m)$.

Next, we show the independence of $\mathbf{G}\mathbf{u}$ and $\mathbf{G}\mathbf{v}$. Equivalently, we are supposed to prove that $\mathbf{e}_i^\top \mathbf{G}\mathbf{u}$ and $\mathbf{e}_{i'}^\top \mathbf{G}\mathbf{v}$ are independent random variables for any $i, i' \in [n]$. For $i \neq i'$, $(\mathbf{e}_i^\top \mathbf{G})\mathbf{u}$ and $(\mathbf{e}_{i'}^\top \mathbf{G})\mathbf{v}$ are independent random variables since $\mathbf{G}$ has independent rows. Therefore, the problem is reduced as the independence of $\mathbf{g}^\top \mathbf{u}$ and $\mathbf{g}^\top \mathbf{v}$ for $\mathbf{g} \sim \mathcal{N}(\mathbf{0}_n, \mathbf{I}_n)$. Notice that

$$[\mathbf{u}, \mathbf{v}]^\top \mathbf{g} \sim \mathcal{N}(\mathbf{0}_2, [\mathbf{u}, \mathbf{v}]^\top \mathbf{I}_n [\mathbf{u}, \mathbf{v}]) = \mathcal{N}(\mathbf{0}_2, \mathbf{I}_2), \tag{24}$$

which completes the proof. $\qquad\square$

Now we are ready to prove the main theorem, which provides the estimate on the rank of (3).

**Theorem 2.** *(A detailed version of Theorem 1) Let the parameters $\mathbf{W}_q, \mathbf{W}_k$ be Gaussian random matrices, i.e., the entries of $\mathbf{W}_q, \mathbf{W}_k$ are independent $\mathcal{N}(0, 1)$ random variables. Assume that the input sequence $\mathbf{X}$ satisfies $\mathbf{X}\mathbf{X}^\top = \mathbf{I}_n$. Then for any $n \in \mathbb{N}_+$, $n \geq 2$, we have*

$$\mathbb{E}_{\mathbf{W}_k, \mathbf{W}_q}\left[ \text{rank}\left( \text{hardmax}\left( \mathbf{X}\mathbf{W}_q \mathbf{W}_k^\top \mathbf{X}^\top \right) \right) \right] \tag{25}$$

$$\leq (1 - \exp(-1))n + 2\exp(-2)[1 + 1/(n-1)]^2 \tag{26}$$

$$\approx (1 - \exp(-1))n \approx 0.63n, \quad n \text{ appropriately large.} \tag{27}$$

*Proof.* According to Lemma 4, since $\mathbf{x}_i^\top \mathbf{x}_j = \delta_{ij}$ (Kronecker symbol), $i, j = 1, 2, \cdots, n$, one can deduce that $\{\mathbf{q}_i\}_{i=1}^n = \{\mathbf{W}_q^\top \mathbf{x}_i\}_{i=1}^n$ is a collection of independent $\mathcal{N}(\mathbf{0}_{d_h}, \mathbf{I}_{d_h})$ random vectors. For any fixed Gaussian random matrix $\mathbf{W}_k$,

$$(\mathbf{e}_i^\top \mathbf{X}\mathbf{W}_q \mathbf{W}_k^\top \mathbf{X}^\top)^\top = \mathbf{K}\mathbf{q}_i \sim \mathcal{N}(\mathbf{0}_n, \mathbf{K}\mathbf{K}^\top), \tag{28}$$

which is also independent across different $i$'s. That is to say, the rows of $\mathbf{X}\mathbf{W}_q\mathbf{W}_k^\top\mathbf{X}^\top$ are independent and identically distributed as $\mathcal{N}(\mathbf{0}_n, \mathbf{K}\mathbf{K}^\top)$. Therefore, according to Lemma 3, the expectation number of columns with all zeros in $\mathrm{hardmax}(\mathbf{X}\mathbf{W}_q\mathbf{W}_k^\top\mathbf{X}^\top)$ is

$$\sum_{j=1}^{n}\prod_{i=1}^{n}(1-p_{ij}) = \sum_{j=1}^{n}\prod_{i=1}^{n}(1-p_j) = \sum_{j=1}^{n}(1-p_j)^n. \tag{29}$$

Hence, we have

$$\frac{1}{n}\mathbb{E}_{\mathbf{W}_q}\left[\mathrm{rank}\left(\mathrm{hardmax}\left(\mathbf{X}\mathbf{W}_q\mathbf{W}_k^\top\mathbf{X}^\top\right)\right)\right] \leq 1 - \frac{1}{n}\sum_{j=1}^{n}(1-p_j)^n. \tag{30}$$

Note that $[p_1, p_2, \cdots, p_n]$ is a probability vector, i.e. $\sum_{j=1}^{n} p_j = 1$, $p_j \geq 0$ for any $j \in [n]$, and $\exp(-p) \geq 1 - p \geq 0$ for any $p \in [0, 1]$, we get $\delta_n(p) = \exp(-pn) - (1-p)^n \geq 0$ for any $p \in [0, 1]$. Therefore, by Lemma 2, we have

$$\frac{1}{n}\sum_{j=1}^{n}|(1-p_j)^n - \exp(-p_jn)| = \frac{1}{n}\sum_{j=1}^{n}\delta_n(p_j) \leq 2\exp(-2)\left(\frac{1}{n-1} + \frac{1}{(n-1)^2}\right), \quad n \geq 2, \tag{31}$$

which gives

$$\frac{1}{n}\sum_{j=1}^{n}(1-p_j)^n = \frac{1}{n}\sum_{j=1}^{n}\exp\left(-p_jn\right) + \frac{1}{n}\sum_{j=1}^{n}\left[(1-p_j)^n - \exp\left(-p_jn\right)\right]$$

$$\geq \left(\prod_{j=1}^{n}\exp\left(-p_jn\right)\right)^{\frac{1}{n}} - 2\exp(-2)\left(\frac{1}{n-1} + \frac{1}{(n-1)^2}\right)$$

$$= \left(\exp\left(-n\sum_{j=1}^{n}p_j\right)\right)^{\frac{1}{n}} - 2\exp(-2)\left(\frac{1}{n-1} + \frac{1}{(n-1)^2}\right)$$

$$= \exp\left(-1\right) - 2\exp(-2)\left(\frac{1}{n-1} + \frac{1}{(n-1)^2}\right), \quad n \geq 2, \tag{32}$$

where the AM-GM inequality is applied, and the equality holds if and only if $p_1 = p_2 = \cdots = p_n$. Hence, the right hand side of (30) $\leq 1 - \exp\left(-1\right) + 2\exp(-2)[1/(n-1) + 1/(n-1)^2]$. Since the estimate holds for any fixed Gaussian random matrix $\mathbf{W}_k$, the proof is completed. $\qquad\square$

## B.1 EXTENSIONS

In this subsection, we extend Theorem 2 to the *almost* orthonormality setting, where the input sequence $\tilde{\mathbf{X}} \in \mathbb{R}^{n \times d}$ satisfies $\tilde{\mathbf{X}}\tilde{\mathbf{X}}^\top = \mathbf{I}_n + \mathbf{E}$, with $\mathbf{E} = [E_{ij}] \in \mathbb{R}^{n \times n}$ satisfying $|E_{ij}| \leq \epsilon \ll 1$ for any $i, j \in [n]$, we adopt the following approximation procedure:

1. Approximate the almost orthonormal input sequence with the exactly orthonormal sequence.
2. Bound the difference between attention products.
3. The desired results follow based on the stability and perturbation analysis.

(i) The first step is to approximate $\tilde{\mathbf{X}}$ with orthonormal matrices:[2]

$$\min_{\mathbf{P} \in \mathbb{R}^{d \times n}: \mathbf{P}^\top\mathbf{P}=\mathbf{I}_n} \|\mathbf{P} - \tilde{\mathbf{X}}^\top\|_F, \tag{33}$$

which can be explicitly solved in a closed form as follows.

---

[2]This is also called the orthogonal procrustes problem (Gower & Dijksterhuis, 2004).

**Lemma 5.** *Assume $d \geq n$. Let $\tilde{\mathbf{X}}^\top = \mathbf{U}\mathbf{\Sigma}\mathbf{V}^\top$ be the singular value decomposition (SVD) of $\tilde{\mathbf{X}}^\top$, where $\mathbf{U} \in \mathbb{R}^{d \times d}$ and $\mathbf{V} \in \mathbb{R}^{n \times n}$ are orthonormal and collect the singular vectors, $\mathbf{\Sigma} = \begin{bmatrix} \mathbf{\Sigma}_r & 0 \\ 0 & 0 \end{bmatrix} \in \mathbb{R}^{d \times n}$ with $\mathbf{\Sigma}_r = \mathrm{diag}(\sigma_1, \sigma_2, \cdots, \sigma_r)$ collecting the singular values ($\sigma_1 \geq \sigma_2 \geq \cdots \geq \sigma_r > 0$, $r = \mathrm{rank}(\tilde{\mathbf{X}}) \leq n$). Then we have*

$$\arg\min_{\mathbf{P} \in \mathbb{R}^{d \times n}: \mathbf{P}^\top \mathbf{P} = \mathbf{I}_n} \|\mathbf{P} - \tilde{\mathbf{X}}^\top\|_F = \mathbf{U}_1 \mathbf{V}^\top, \tag{34}$$

*where $\mathbf{U}_1 := \mathbf{U}\begin{bmatrix} \mathbf{I}_n \\ 0 \end{bmatrix} \in \mathbb{R}^{d \times n}$ denotes the first $n$ columns of $\mathbf{U}$. Furthermore, if the input sequence $\tilde{\mathbf{X}} \in \mathbb{R}^{n \times d}$ is almost orthonormal such that $\tilde{\mathbf{X}}\tilde{\mathbf{X}}^\top = \mathbf{I}_n + \mathbf{E}$ with $\mathbf{E} = [E_{ij}] \in \mathbb{R}^{n \times n}$ satisfying $|\mathbf{E}_{ij}| \leq \epsilon = o(1/n^{\frac{3}{2}})$ ($\forall i, j \in [n]$), then $r = \mathrm{rank}(\tilde{\mathbf{X}}) = n$, and we have the following estimate*

$$\|\mathbf{U}_1 \mathbf{V}^\top - \tilde{\mathbf{X}}^\top\|_F \leq \epsilon n^{\frac{3}{2}} = o(1). \tag{35}$$

*Proof.* First, we can derive that

$$\arg\min_{\mathbf{P} \in \mathbb{R}^{d \times n}: \mathbf{P}^\top \mathbf{P} = \mathbf{I}_n} \|\mathbf{P} - \tilde{\mathbf{X}}^\top\|_F^2 = \arg\min_{\mathbf{P} \in \mathbb{R}^{d \times n}: \mathbf{P}^\top \mathbf{P} = \mathbf{I}_n} \mathrm{trace}((\mathbf{P} - \tilde{\mathbf{X}}^\top)^\top (\mathbf{P} - \tilde{\mathbf{X}}^\top))$$

$$= \arg\min_{\mathbf{P} \in \mathbb{R}^{d \times n}: \mathbf{P}^\top \mathbf{P} = \mathbf{I}_n} \mathrm{trace}(\mathbf{P}^\top \mathbf{P} - \mathbf{P}^\top \tilde{\mathbf{X}}^\top - \tilde{\mathbf{X}}\mathbf{P} + \tilde{\mathbf{X}}\tilde{\mathbf{X}}^\top)$$

$$= \arg\max_{\mathbf{P} \in \mathbb{R}^{d \times n}: \mathbf{P}^\top \mathbf{P} = \mathbf{I}_n} \mathrm{trace}(\tilde{\mathbf{X}}\mathbf{P})$$

$$= \arg\max_{\mathbf{P} \in \mathbb{R}^{d \times n}: \mathbf{P}^\top \mathbf{P} = \mathbf{I}_n} \mathrm{trace}(\mathbf{\Sigma}^\top \cdot \mathbf{U}^\top \mathbf{P}\mathbf{V}). \tag{36}$$

Let $\mathbf{S} := \mathbf{U}^\top \mathbf{P}\mathbf{V} = [S_{ij}] \in \mathbb{R}^{d \times n}$, then $\mathbf{S}^\top \mathbf{S} = \mathbf{V}^\top \mathbf{P}^\top \mathbf{U}\mathbf{U}^\top \mathbf{P}\mathbf{V} = \mathbf{I}_n$, which yields $1 = \sum_{j=1}^d S_{ji}^2 \geq S_{ii}^2$ for any $i \in [n]$. Therefore, note that

$$\mathrm{trace}(\mathbf{\Sigma}^\top \cdot \mathbf{S}) = \sum_{i=1}^r \sigma_i S_{ii} \leq \sum_{i=1}^r \sigma_i |S_{ii}| \leq \sum_{i=1}^r \sigma_i, \tag{37}$$

and the equality holds when $S_{ii} = 1$ for any $i \in [r]$, we deduce that

$$\arg\max_{\mathbf{S} \in \mathbb{R}^{d \times n}: \mathbf{S}^\top \mathbf{S} = \mathbf{I}_n} \mathrm{trace}(\mathbf{\Sigma}^\top \cdot \mathbf{S}) = \begin{bmatrix} \mathbf{I}_n \\ 0 \end{bmatrix}. \tag{38}$$

Combining with (36), we equivalently obtain

$$\arg\min_{\mathbf{P} \in \mathbb{R}^{d \times n}: \mathbf{P}^\top \mathbf{P} = \mathbf{I}_n} \|\mathbf{P} - \tilde{\mathbf{X}}^\top\|_F^2 = \arg\max_{\mathbf{P} \in \mathbb{R}^{d \times n}: \mathbf{P}^\top \mathbf{P} = \mathbf{I}_n} \mathrm{trace}(\mathbf{\Sigma}^\top \cdot \mathbf{U}^\top \mathbf{P}\mathbf{V})$$

$$= \mathbf{U}\begin{bmatrix} \mathbf{I}_n \\ 0 \end{bmatrix} \mathbf{V}^\top = \mathbf{U}_1 \mathbf{V}^\top, \tag{39}$$

which proves (34). To prove (35), note that $\sigma_i^2$ is the $i$-th eigenvalue of $\tilde{\mathbf{X}}\tilde{\mathbf{X}}^\top$, according to Weyl's theorem, we have

$$|\sigma_i^2 - 1| \leq \|\tilde{\mathbf{X}}\tilde{\mathbf{X}}^\top - \mathbf{I}_n\|_2 = \|\mathbf{E}\|_2, \quad i \in [n]. \tag{40}$$

Since

$$\|\mathbf{E}\|_2^2 = \max_{\mathbf{z} \in \mathbb{R}^n: \|\mathbf{z}\|_2 = 1} \|\mathbf{E}\mathbf{z}\|_2^2 = \max_{\mathbf{z} \in \mathbb{R}^n: \|\mathbf{z}\|_2 = 1} \sum_{i=1}^n |\mathbf{E}_{i,:} \cdot \mathbf{z}|^2 \tag{41}$$

$$\leq \max_{\mathbf{z} \in \mathbb{R}^n: \|\mathbf{z}\|_2 = 1} \sum_{i=1}^n \|\mathbf{E}_{i,:}\|_2^2 \|\mathbf{z}\|_2^2 = \|\mathbf{E}\|_F^2 \leq \epsilon^2 n^2, \tag{42}$$

where $\mathbf{E}_{i,:}$ denotes the $i$-th row of $\mathbf{E}$, we get

$$|\sigma_i^2 - 1| \leq \epsilon n = o(1/\sqrt{n}), \quad i \in [n], \tag{43}$$

leading to $\sigma_i > 0$ for any $i \in [n]$, and hence $\tilde{\mathbf{X}}$ has the full rank $r = \text{rank}(\tilde{\mathbf{X}}) = n$. Therefore

$$\|\mathbf{U}_1\mathbf{V}^\top - \tilde{\mathbf{X}}^\top\|_F^2 = \left\|\mathbf{U}\begin{bmatrix}\mathbf{I}_n\\\mathbf{0}\end{bmatrix}\mathbf{V}^\top - \mathbf{U}\mathbf{\Sigma}\mathbf{V}^\top\right\|_F^2 = \left\|\begin{bmatrix}\mathbf{I}_n\\\mathbf{0}\end{bmatrix} - \begin{bmatrix}\mathbf{\Sigma}_n\\\mathbf{0}\end{bmatrix}\right\|_F^2$$

$$= \sum_{i=1}^n |1 - \sigma_i|^2 = \sum_{i=1}^n \frac{|1 - \sigma_i^2|^2}{|1 + \sigma_i|^2} \leq \sum_{i=1}^n \epsilon^2 n^2 = \epsilon^2 n^3 = o(1), \qquad (44)$$

which completes the proof. $\qquad\square$

(ii) As the second step, the difference between attention products can be further bounded as follows.

**Lemma 6.** *Let $\mathbf{X} := \mathbf{V}\mathbf{U}_1^\top$ with $\mathbf{V}, \mathbf{U}_1$ defined in Lemma 5. Under the same conditions in Lemma 5, and further assume $\epsilon = o(1/(n^{\frac{3}{2}}(d + d_h)))$ we have the following estimates:*

*1. For any $t > 0$, with probability at least $(1 - 2\exp(-t^2))^2$, it holds that*

$$\|\mathbf{X}\mathbf{W}_q\mathbf{W}_k^\top\mathbf{X}^\top - \tilde{\mathbf{X}}\mathbf{W}_q\mathbf{W}_k^\top\tilde{\mathbf{X}}^\top\|_2 \lesssim \epsilon n^{\frac{3}{2}}(d + d_h + t^2) = o(1). \qquad (45)$$

*2. $\mathbb{E}_{\mathbf{W}_k,\mathbf{W}_q}\|\mathbf{X}\mathbf{W}_q\mathbf{W}_k^\top\mathbf{X}^\top - \tilde{\mathbf{X}}\mathbf{W}_q\mathbf{W}_k^\top\tilde{\mathbf{X}}^\top\|_2 \lesssim \epsilon n^{\frac{3}{2}}(d + d_h) = o(1)$.*

*Here, $\lesssim$ hides positive absolute constants.*

*Proof.* Let $\mathbf{P} := \tilde{\mathbf{X}} - \mathbf{X}$. According to Lemma 5, we have $\|\mathbf{P}\|_F \leq \epsilon n^{\frac{3}{2}} = o(1)$. Then, we can derive that

$$\|\mathbf{X}\mathbf{W}_q\mathbf{W}_k^\top\mathbf{X}^\top - \tilde{\mathbf{X}}\mathbf{W}_q\mathbf{W}_k^\top\tilde{\mathbf{X}}^\top\|_2 = \|\mathbf{X}\mathbf{W}_q\mathbf{W}_k^\top\mathbf{X}^\top - (\mathbf{X} + \mathbf{P})\mathbf{W}_q\mathbf{W}_k^\top(\mathbf{X} + \mathbf{P})^\top\|_2$$

$$= \|\mathbf{P}\mathbf{W}_q\mathbf{W}_k^\top\mathbf{X}^\top + \mathbf{X}\mathbf{W}_q\mathbf{W}_k^\top\mathbf{P}^\top + \mathbf{P}\mathbf{W}_q\mathbf{W}_k^\top\mathbf{P}^\top\|_2$$

$$\leq 2\|\mathbf{P}\|_2\|\mathbf{W}_q\|_2\|\mathbf{W}_k\|_2\|\mathbf{X}\|_2 + \|\mathbf{P}\|_2^2\|\mathbf{W}_q\|_2\|\mathbf{W}_k\|_2. \quad (46)$$

Note that $\|\mathbf{P}\|_2 \leq \|\mathbf{P}\|_F \leq \epsilon n^{\frac{3}{2}} = o(1)$, $\|\mathbf{X}\|_2 = \|\mathbf{U}_1\|_2 = \|\mathbf{I}_n\|_2 = 1$, the remaining task is to estimate $\|\mathbf{W}\|_2$ for any Gaussian random matrix $\mathbf{W}$ (i.e., the entries of $\mathbf{W}$ are independent $\mathcal{N}(0,1)$ random variables). According to Vershynin (2018) (Theorem 4.4.5, Exercise 4.4.6 and Example 2.5.8), we have for any $t > 0$,

$$\|\mathbf{W}\|_2 \lesssim \sqrt{d} + \sqrt{d_h} + t, \quad \text{with probability at least } 1 - 2\exp(-t^2), \qquad (47)$$

where $\lesssim$ hides positive absolute constants, and

$$\mathbb{E}\|\mathbf{W}\|_2 \lesssim \sqrt{d} + \sqrt{d_h}. \qquad (48)$$

Combining with (46), we have for any $t > 0$,

$$\|\mathbf{X}\mathbf{W}_q\mathbf{W}_k^\top\mathbf{X}^\top - \tilde{\mathbf{X}}\mathbf{W}_q\mathbf{W}_k^\top\tilde{\mathbf{X}}^\top\|_2$$

$$\leq 2\|\mathbf{P}\|_2\|\mathbf{W}_q\|_2\|\mathbf{W}_k\|_2\|\mathbf{X}\|_2 + \|\mathbf{P}\|_2^2\|\mathbf{W}_q\|_2\|\mathbf{W}_k\|_2$$

$$\lesssim (\epsilon n^{\frac{3}{2}} + \epsilon^2 n^3)(\sqrt{d} + \sqrt{d_h} + t)^2$$

$$\lesssim \epsilon n^{\frac{3}{2}}(d + d_h + t^2) = o(1), \quad \text{with probability at least } (1 - 2\exp(-t^2))^2, \qquad (49)$$

and

$$\mathbb{E}_{\mathbf{W}_k,\mathbf{W}_q}\|\mathbf{X}\mathbf{W}_q\mathbf{W}_k^\top\mathbf{X}^\top - \tilde{\mathbf{X}}\mathbf{W}_q\mathbf{W}_k^\top\tilde{\mathbf{X}}^\top\|_2$$

$$\leq 2\|\mathbf{P}\|_2\|\mathbf{X}\|_2 \cdot \mathbb{E}_{\mathbf{W}_q}\|\mathbf{W}_q\|_2 \cdot \mathbb{E}_{\mathbf{W}_k}\|\mathbf{W}_k\|_2 + \|\mathbf{P}\|_2^2 \cdot \mathbb{E}_{\mathbf{W}_q}\|\mathbf{W}_q\|_2 \cdot \mathbb{E}_{\mathbf{W}_k}\|\mathbf{W}_k\|_2$$

$$\lesssim (\epsilon n^{\frac{3}{2}} + \epsilon^2 n^3)(\sqrt{d} + \sqrt{d_h})^2 \lesssim \epsilon n^{\frac{3}{2}}(d + d_h) = o(1), \qquad (50)$$

which completes the proof. $\qquad\square$

(iii) The third step is to apply the stability and perturbation analysis.

**Proposition 1.** *(Stability of numerical ranks) Let $\sigma_{min} \neq 0$ denote the minimal non-zero singular value of a matrix $\mathbf{A}$. Then for any perturbation $\mathbf{P}$ with $\|\mathbf{P}\|_2 \leq \sigma_{min}/3$ and any $\delta \in (\sigma_{min}/3, 2\sigma_{min}/3]$, we have*

$$\operatorname{rank}(\mathbf{A}, \delta) = \operatorname{rank}(\mathbf{A} + \mathbf{P}, \delta). \tag{51}$$

*Proof.* By definition, the numerical rank $\operatorname{rank}(\mathbf{A}, \delta)$ equals to the number of singular values (of $\mathbf{A}$) no less than $\delta$. Therefore, for any $\delta \in (0, \sigma_{\min}]$, $\operatorname{rank}(\mathbf{A}, \delta)$ equals to the number of non-zero singular values of $\mathbf{A}$. Let $\{\sigma_i\}$ and $\{\tilde{\sigma}_i\}$ be the singular values of $\mathbf{A}$ and $\mathbf{A} + \mathbf{P}$, respectively. According to Weyl's theorem, we have $|\sigma_i - \tilde{\sigma}_i| \leq \|\mathbf{P}\|_2 \leq \sigma_{\min}/3$. Then for any $\delta \in (\sigma_{\min}/3, 2\sigma_{\min}/3]$, the perturbation of non-zero singular values satisfies $\tilde{\sigma}_i \geq \sigma_i - \sigma_{\min}/3 \geq \sigma_{\min} - \sigma_{\min}/3 \geq \delta$, which is selected for counting the numerical rank, and the perturbation of zero singular values satisfies $\tilde{\sigma}_i \leq \sigma_{\min}/3 < \delta$, which is not selected for counting the numerical rank. That is, $\operatorname{rank}(\mathbf{A} + \mathbf{P}, \delta)$ still equals to the number of non-zero singular values of $\mathbf{A}$, hence the desired result follows. $\square$

**Further perturbation analysis**  The subsequent analysis is similar, since all the remaining operations (activation, numerical rank and expectation) are *stable*. In fact, both the activation and expectation are continuous with respect to perturbations of inputs, and so does the numerical rank due to Proposition 1. Therefore, the derived upper bounds in Theorem 1 or Theorem 2 still hold for almost orthonormal input sequences.

## C FURTHER DETAILS OF ABLATION STUDIES

### C.1 EFFECT OF MODEL DIMENSIONS

In this section, we study the effect of model dimensions on the attention rank of Transformers. We test for different dimensions $d_{\text{model}} \in \{384, 768, 1152, 1536\}$, maintaining other configurations specified in Section 3.1. The results illustrated in Figure 5 align with the phenomena observed in Figure 1 and Table 1, indicating a robust and consistent pattern of attention ranks across varied model dimensions.

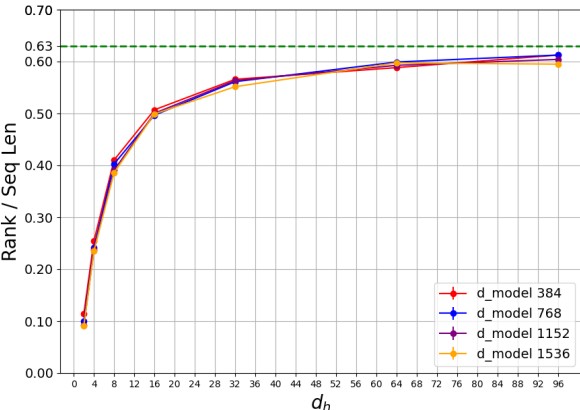

Figure 5: The attention ranks across different model dimensions.

### C.2 EFFECT OF SOFTMAX TEMPERATURES

In this section, we investigate the impact of softmax temperatures on the attention rank of Transformer models. We test for different temperatures $T \in \{10^{-5}, 10^{-3}, 10^{-1}, 1\}$, and all the other configurations remain the same as those of Section 3.1.

The softmax temperature is an important factor that influences the sharpness of the attention distribution. Lower temperatures lead to more concentrated attention distributions, effectively pushing the softmax activation towards the hardmax activation. Conversely, higher temperatures yield more uniform attention distributions. Despite of these differences, our results show consistent patterns of attention ranks across all tested temperatures. This consistency, as is depicted in Figure 6, suggests that the attention rank of Transformers is robust to variations in softmax temperatures.

### C.3 EFFECT OF TRANSFORMERS' LAYERS

In this section, we detail the influence of Transformers' layers on the attention rank. The experiment utilizes a model configuration with 8 layers to examine the attention rank's behavior across layers, and the other configurations are consistent with Section 3.1.

The results shown in Figure 7 exhibit a noticeable trend: with the increase of depth, the attention mechanism tends to show a more pronounced low-rank behavior. This trend is particularly evident in the deeper layers of the Transformer, suggesting that the model depth significantly influences the dynamics of attention ranks.

### C.4 EFFECT OF DATA DISTRIBUTIONS

For a comprehensive analysis of the impact of data distributions on the attention rank of Transformers, we numerically study a range of data distributions including normal distributions $\mathcal{N}(0, 1)$ and $\mathcal{N}(0, 100)$, as well as uniform distributions $\mathcal{U}(-1, 1)$ and $\mathcal{U}(-100, 100)$. These distributions are

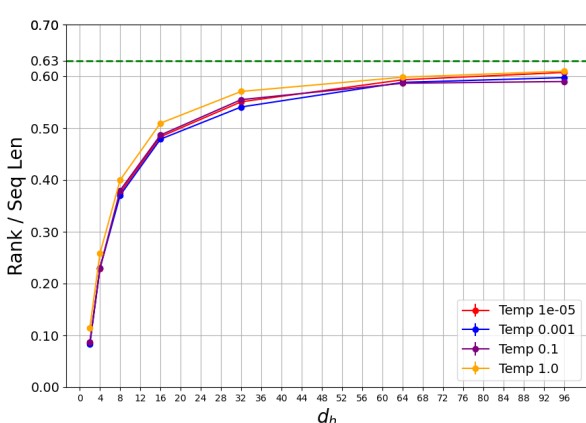

Figure 6: Attention ranks across various softmax temperatures.

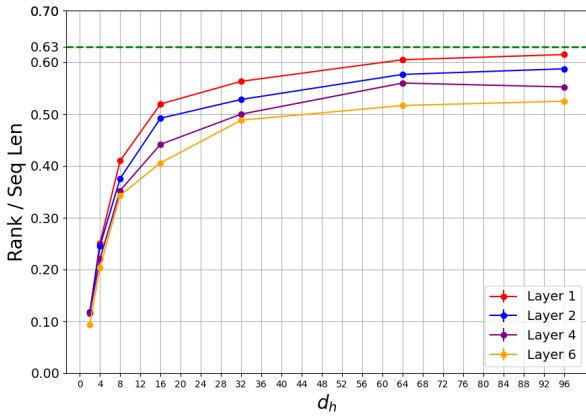

Figure 7: Attention ranks across different Transformer layers.

selected to mimic common scenarios in NLP applications, where input tokens are typically embedded using Gaussian distributions. The model configurations used in these experiments are consistent with Section 3.1. Our findings reveal the remarkable robustness of the attention rank with respect to data distributions, as is evidenced by consistent patterns of attention ranks across all tested data distributions in Table 3. It is particularly notable for the normal distributions ($\mathcal{N}(0, 1)$ and $\mathcal{N}(0, 100)$), which show similar patterns of attention ranks and imply that the initial Gaussian embeddings of input tokens do not significantly influence the attention mechanism's efficacy. The uniform distributions $\mathcal{U}(-1, 1)$ and $\mathcal{U}(-100, 100)$ follow the same trend, reinforcing the model's insensitivity to the nature of data distributions. These results underscore the robustness of Transformer models to variations in data distributions, which is a crucial factor for real-world applications.

### C.5 Numerical Verifications on the Orthonormality

## D Further Details on Real-World Experiments

### D.1 Additional Verifications on Low-Rank Barrier

**Additional datasets.** In this section, we present supplementary results on the performance of Vision Transformers (ViTs) under varied model dimensions on the CIFAR-10, CIFAR-100 and SVHN dataset. In these experiments, we maintain the relationship $d = d_{\text{model}} = h \times d_h$. These results fur-

Table 3: The attention ranks for different data distributions: $\mathcal{N}(0,1)$, $\mathcal{N}(0,100)$, $\mathcal{U}(-1,1)$ and $\mathcal{U}(-100,100)$. Note that the normal distributions correspond with the practical NLP applications where input tokens are initially embedded with Gaussian distributions. Here, $d_h$ represents the head dimension. The "Rank / Seq Len" is the ratio of attention ranks over sequence lengths, with the standard deviation denoted by $\pm$. The "Improvement" column summarizes the successive increases in the "Rank / Seq Len" column compared to the previous row.

| $\mathcal{N}(0,1)$ | | | $\mathcal{N}(0,100)$ | | | $\mathcal{U}(-1,1)$ | | | $\mathcal{U}(-100,100)$ | | |
|---|---|---|---|---|---|---|---|---|---|---|---|
| $d_h$ | Rank / Seq Len | Improvement | $d_h$ | Rank / Seq Len | Improvement | $d_h$ | Rank / Seq Len | Improvement | $d_h$ | Rank / Seq Len | Improvement |
| 2 | $0.11 \pm 0.023$ | - | 2 | $0.10 \pm 0.014$ | - | 2 | $0.17 \pm 0.039$ | - | 2 | $0.09 \pm 0.016$ | - |
| 4 | $0.25 \pm 0.032$ | +0.14 | 4 | $0.23 \pm 0.029$ | +0.12 | 4 | $0.30 \pm 0.038$ | +0.13 | 4 | $0.23 \pm 0.027$ | +0.14 |
| 8 | $0.40 \pm 0.035$ | +0.15 | 8 | $0.41 \pm 0.034$ | +0.18 | 8 | $0.45 \pm 0.036$ | +0.15 | 8 | $0.38 \pm 0.028$ | +0.15 |
| 16 | $0.51 \pm 0.033$ | +0.11 | 16 | $0.52 \pm 0.036$ | +0.11 | 16 | $0.56 \pm 0.033$ | +0.11 | 16 | $0.49 \pm 0.035$ | +0.11 |
| 32 | $0.57 \pm 0.033$ | +0.06 | 32 | $0.57 \pm 0.038$ | +0.05 | 32 | $0.63 \pm 0.028$ | +0.07 | 32 | $0.56 \pm 0.031$ | +0.07 |
| 64 | $0.60 \pm 0.032$ | +0.03 | 64 | $0.61 \pm 0.032$ | +0.04 | 64 | $0.64 \pm 0.028$ | +0.01 | 64 | $0.59 \pm 0.012$ | +0.03 |
| 96 | $0.61 \pm 0.036$ | +0.01 | 96 | $0.61 \pm 0.018$ | +0.00 | 96 | $0.64 \pm 0.008$ | +0.00 | 96 | $0.60 \pm 0.050$ | +0.01 |

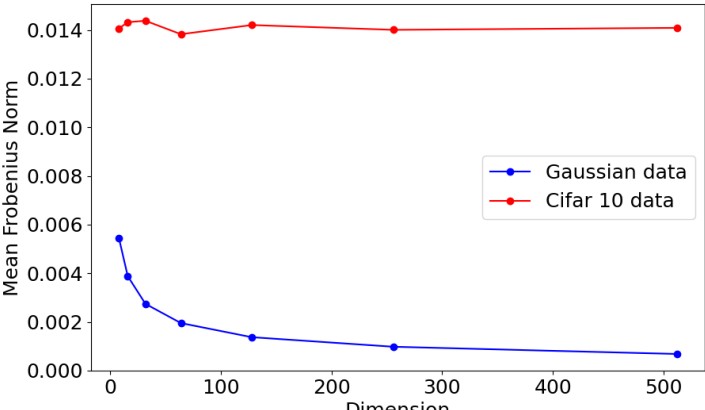

Figure 8: The orthogonality measure across different dimensions for Gaussian random and CIFAR-10 data (after an initialized embedding layer). Here, we use the mean Frobenius norm as the orthogonality measure for tensors with various dimensions. The x-axis represents the (head) dimension $d_h$ (ranging from 8 to 512), while the y-axis indicates the mean Frobenius norm: $\frac{1}{n^2}\|Q - I\|_F$, where $n$ is the sequence length, $Q$ denotes the cosine similarity matrix, and $I$ is the identity matrix. Certainly, lower mean Frobenius norms lead to more orthonormal tokens in the tensor. We observe that both Gaussian random data and CIFAR-10 data exhibit relatively small mean Frobenius norms, indicating that they are nearly orthonormal.

ther corroborate and align with the findings discussed in the main text, demonstrating the existence of the low-rank barrier.

**Final accuracy.** To further demonstrate the impact of the low-rank barrier, we also summarize the final accuracy achieved by each experiment. These results indicate that with the constraint $d = d_{\text{model}} = h \times d_h$, a smaller number of heads $h$ results in a larger head dimension $d_h$, potentially exceeding the necessary head dimensions to approach the low-rank barrier (i.e. exceeding the critical point where the attention rank gets saturated) for each head. Equivalently, most of heads may have reached the low-rank barrier, leading to the parameter redundancy. However, as the number of heads increases, the Transformer model avoids the potential parameter redundancy and obtains more "effective" ranks for modeling, hence yields improved experimental results.

## D.2 ADDITIONAL VERIFICATIONS ON MODEL-REDUCTION EFFECT

In this section, we present a detailed set of experimental results to elucidate the model-reduction effect on various datasets under different configurations. Here, we do not maintain the constraint $d = h \times d_h$, but fix the number of heads as $h = 4, 8$ and vary the head dimension $d_h$ (and hence the model dimension $d_{\text{model}} \neq d$). Notably, although the initial improvement in the validation ac-

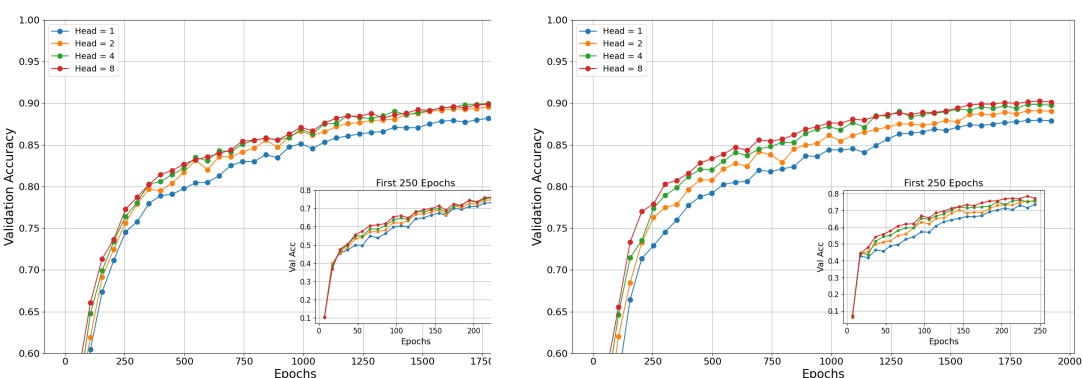

Figure 9: The validation accuracy of ViTs on the CIFAR-10 dataset with the model dimensions 192 (left) and 384 (right).

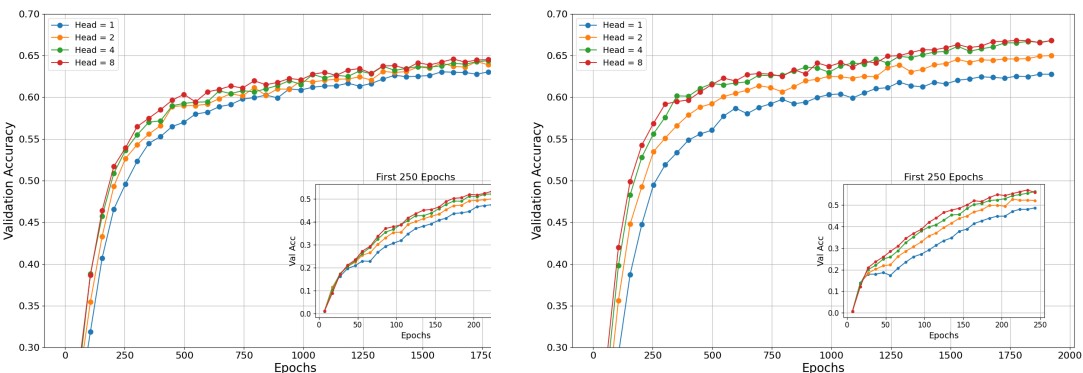

Figure 10: The validation accuracy of ViTs on the CIFAR-100 dataset with the model dimensions 192 (left) and 384 (right).

curacy is pronounced as the head dimension $d_h$ increases within relatively small values, this improvement plateaus for appropriately large values of $d_h$, indicating diminishing returns with further increments in model parameters. These observations align with our theoretical justifications on the model-reduction effect, suggesting an optimal range for head dimensions that balance the model performance with parameter efficiency.

### D.3 ADDITIONAL EXPERIMENTS ON TEXT CLASSIFICATION TASKS

This section provides a detailed examination of the experimental results illustrating the model-reduction effect on the IMDB dataset for text classification tasks. Notably, we deviate from the conventional constraint $d = h \times d_h$ by fixing the number of heads as $h = 1$ while varying the head dimension $d_h$. Consequently, the model dimension $d_{\mathrm{model}} \neq d$. The results are presented in Figure 15. Consistent with the phenomena from image tasks, the validation accuracy on text classification tasks increases significantly as the head dimension $d_h$ grows within a relatively small range; however, this improvement plateaus once $d_h$ becomes appropriately large, reflecting diminishing returns from further expansions in model parameters (Figure 15, left). Also, the attention rank appears *aligned* "plateauing" dynamics with the same critical point of saturation (Figure 15, right), i.e., both the performance gains and attention ranks get saturated at around $d_h^* = 8$. These results underscore the presence of optimal ranges for head dimensions that balance performance gains and

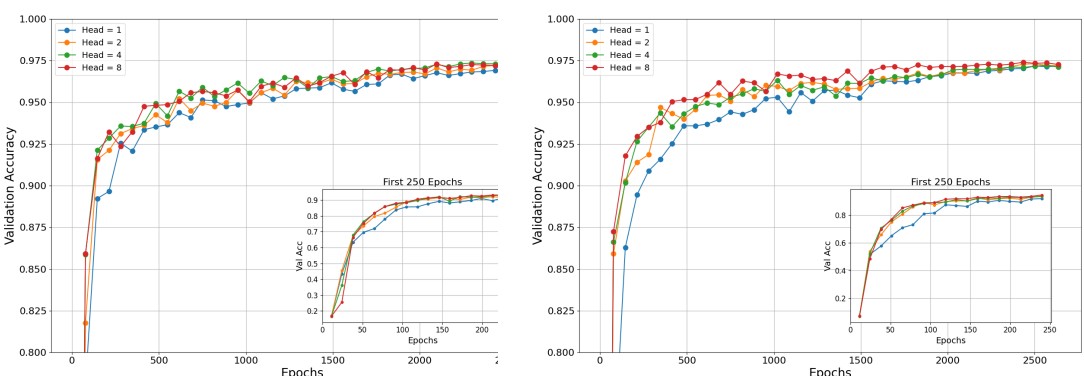

Figure 11: The validation accuracy of ViTs on the SVHN dataset with the model dimensions 192 (left) and 384 (right).

Table 4: The final accuracy for different models on varied datasets.

| Configurations | | Final accuracy | | | | |
|---|---|---|---|---|---|---|
| Datasets | $d_{\mathrm{model}}$ | Head = 1 | Head = 2 | Head = 4 | Head = 8 | Head = 16 |
| Cifar-10 | 192 | 0.8836 | 0.8981 | 0.9004 | 0.9013 | 0.8932 |
| Cifar-10 | 384 | 0.8795 | 0.8924 | 0.8977 | 0.9000 | 0.8997 |
| Cifar-100 | 192 | 0.6316 | 0.6435 | 0.6454 | 0.6470 | 0.6378 |
| Cifar-100 | 384 | 0.6280 | 0.6497 | 0.6685 | 0.6680 | 0.6671 |
| SVHN | 192 | 0.9684 | 0.9717 | 0.9737 | 0.9739 | 0.9724 |
| SVHN | 384 | 0.9721 | 0.9723 | 0.9713 | 0.9730 | 0.9757 |

parameter efficiency effectively. Furthermore, to study the effect of input sizes on attention ranks, we also test for different values of (input) embedding dimensions within $\{32, 128, 256, 512\}$ on the IMDB dataset. The experimental results are shown in Figure 16. It is similarly observed that the rank saturation phenomenon still appears as the input size varies.

**Broader Impacts**  This paper presents studies with the goal to advance the field of machine learning. There are many potential societal consequences of our work, none of which we feel must be specifically highlighted here. As far as we know, our paper has no potential negative societal impacts.

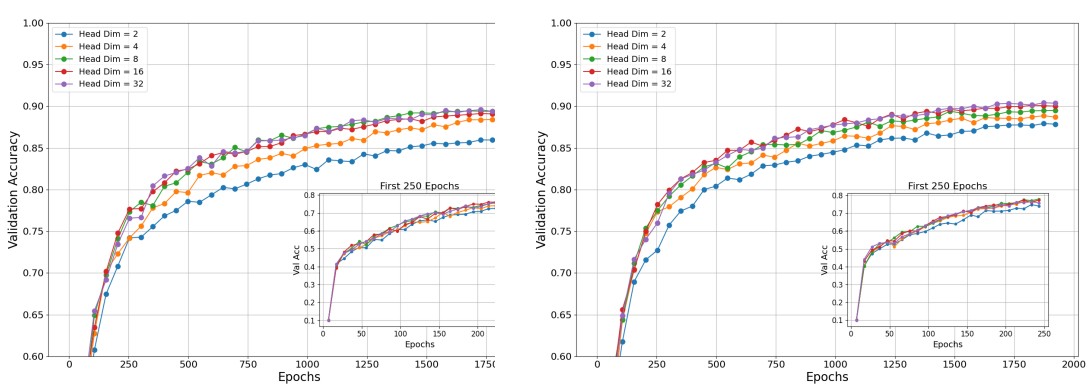

Figure 12: The validation accuracy of ViTs on the CIFAR-10 dataset with 4 heads (left) and 8 heads (right).

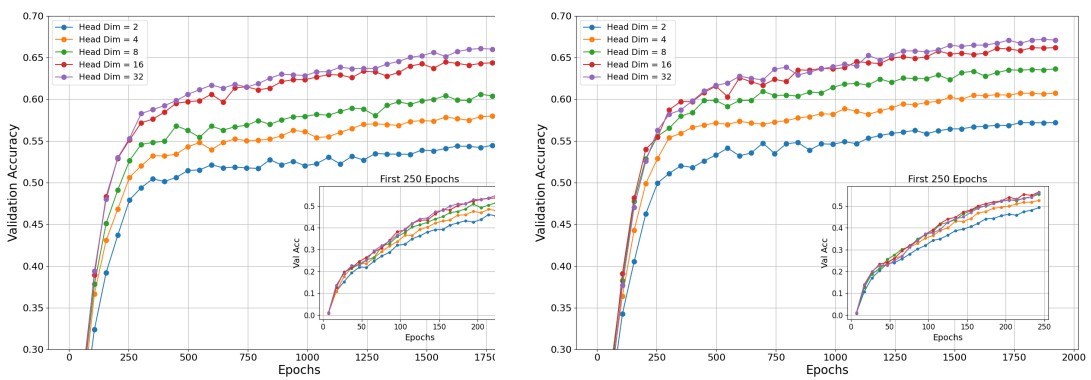

Figure 13: The validation accuracy of ViTs on the CIFAR-100 dataset with 4 heads (left) and 8 heads (right).

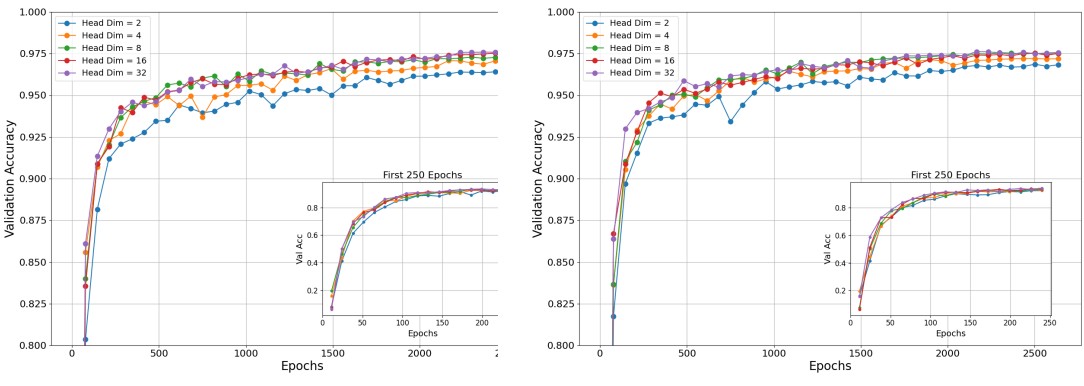

Figure 14: The validation accuracy of ViTs on the SVHN dataset with 4 heads (left) and 8 heads (right).

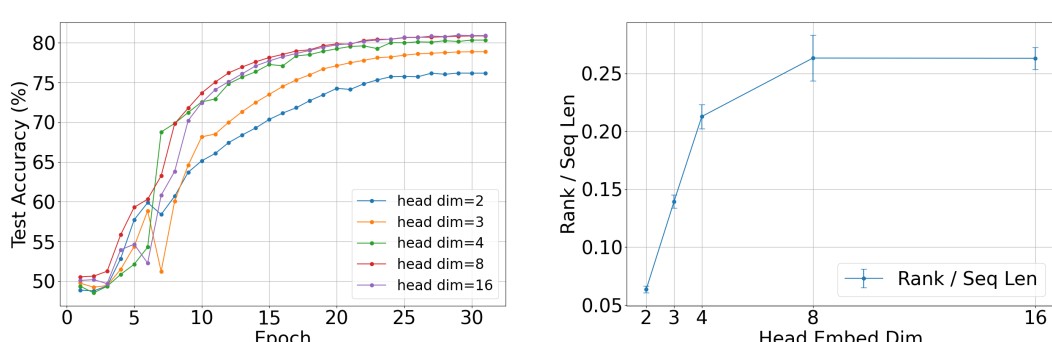

Figure 15: Experimental results of text classification tasks on the IMDB dataset. Left: Learning accuracies of different head dimensions along the training. Right: Attention ranks corresponding to the first-layer attention matrices, computed on mini-batches of IMDB tokens and averaged over multiple runs using varied random seeds. Here, five distinct head dimensions are evaluated: $d_h = 2, 3, 4, 8, 16$. The observed pattern of attention ranks aligns with Figure 1, where smaller values of $d_h$ result in notable increases in attention ranks as $d_h$ grows; however, when $d_h \geq 8$, further increases in $d_h$ lead to marginal changes in attention ranks. Importantly, the trends in attention ranks closely parallel the trends of model performance, which is consistent with the image setting (Figure 4). In fact, both attention ranks and model performance improve with increasing the head dimension $d_h$ but plateau at $d_h \geq d_h^* = 8$, indicating $d_h^*$ as the optimal configuration to trade-off between model efficiency and learning performance.

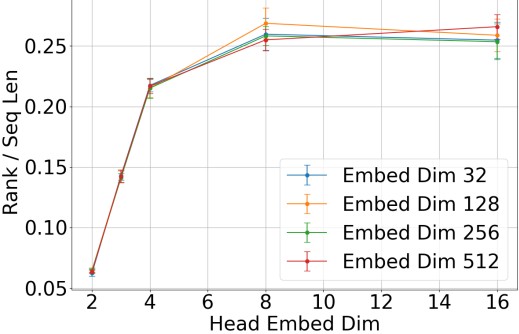

Figure 16: Rank saturation phenomenon across different input embedding dimensions on the IMDB dataset.

