# OpenReview forum: "On the Limitation and Redundancy of Transformers: A Rank Perspective"
_ICLR.cc/2025/Conference — Submitted to ICLR 2025_

### Official Review · Reviewer_VcLz · 2024-10-27

**Soundness:** 4
**Presentation:** 3
**Contribution:** 3
**Rating:** 8
**Confidence:** 4

**Summary:**

The paper studies the limitations of the rank of attention matrices, with a goal of gaining a better understanding of the expressive power of transformers. First, the paper provides experiments on randomly generated data from different distributions and show empirically a rank saturation at around 0.63n (n being the input dimension). Next, it is proved theoretically that transformers at initialization also exhibit this rank saturation. Finally, experiments are given on the CIFAR-10/100 and SVHN datasets showing a rank saturation phenomenon.

**Strengths:**

The phenomenon that is presented in the paper is interesting and sheds new light on the expressive power of attention matrices. The paper provides theoretical results and complementing empirical validation under different setting. The technical parts of the paper are clearly written and easy to follow.

**Weaknesses:**

I have some concerns with the presentation of the paper and the theoretical and empirical results:

1) The results of Sections 3 and 4 discuss transformers with random data, random weights, or both. It is difficult to generalize from that to general transformers trained on real data. In general, I believe it is OK to have a theoretical paper with restricted assumptions. However, the presentation of this work, and specifically the abstract and introduction, gives the impression that the low-rank phenomenon is general and not restricted to random settings. If the focus of the paper is random settings (i.e. random data and/or weights) this should be clearly stated. If the message of the paper is more general, then further evidence should be provided as I discuss later.

2) The theoretical result (Theorem 1) is nice, but limited and isn’t enough in itself for a theoretical paper. The biggest limitation is the assumption of the data. Although the authors justify assuming that the samples are almost orthogonal (e.g. drawn from a Gaussian or uniform on a sphere), the assumption is that they are exactly orthogonal. This allows only for n samples, instead of O(2^n) samples. It seems possible to prove this result for almost orthogonal data.

3) The “real-world experiments” in section 5 are done on very small-scale image datasets, CIFAR-10/100 and SVHN. It would be more convincing to do experiments on larger datasets, and specifically text datasets where it is possible to change the embedding dimension, and thus experiment on the effects of changing n.

**Questions:**

- What would the effect of the rank in Theorem 1 be when having data that is almost orthogonal?

- Does the rank saturation phenomenon also happen in real datasets when the input dimension n varies?

---

> ### Author Response · Authors · 2024-11-27
> **Response to Reviewer VcLz**
>
> >**Q1**: The results of Sections 3 and 4 discuss transformers with random data, random weights, or both. It is difficult to generalize from that to general transformers trained on real data. In general, I believe it is OK to have a theoretical paper with restricted assumptions. However, the presentation of this work, and specifically the abstract and introduction, gives the impression that the low-rank phenomenon is general and not restricted to random settings. If the focus of the paper is random settings (i.e. random data and/or weights) this should be clearly stated. If the message of the paper is more general, then further evidence should be provided as I discuss later.
>
> **A1**: We clarify that although the theoretical results are developed under random settings, the obtained insights (mainly model-reduction effect) generalize to real-world settings. This point has been discussed in the first paragraph of Section 5 (Line 419-427) in the original manuscript, and we have updated corresponding contents for further clarifications in the revised version.
>
> >**Q2**: The theoretical result (Theorem 1) is nice, but limited and isn’t enough in itself for a theoretical paper. The biggest limitation is the assumption of the data. Although the authors justify assuming that the samples are almost orthogonal (e.g. drawn from a Gaussian or uniform on a sphere), the assumption is that they are exactly orthogonal. This allows only for n samples, instead of O(2^n) samples. It seems possible to prove this result for almost orthogonal data.
>
> **A2**: Thanks for your suggestion. It is sharp to note that almost orthonormality leads to exponentially many data samples (rather than linear for exact orthonormality) due to Johnson–Lindenstrauss lemma. We have successfully extended the main theorem to the almost orthonormality setting via approximation procedures and stability/perturbation analysis. See details in ***Section B.1*** in the revised manuscript.
>
> >**Q3**: The “real-world experiments” in section 5 are done on very small-scale image datasets, CIFAR-10/100 and SVHN. It would be more convincing to do experiments on larger datasets, and specifically text datasets where it is possible to change the embedding dimension, and thus experiment on the effects of changing $n$.
>
> **A3**: Thanks for your suggestion. We have added required NLP experiments (on the IMDB dataset) and discussions in ***Section D.3 (Figure 15 and 16)*** in the revised manuscript. It turns out that the insights and implications obtained under image settings consistently hold under text settings with varied input sizes.
>
> >**Q4**: What would the effect of the rank in Theorem 1 be when having data that is almost orthogonal?
>
> **A4**: Please refer to **A2** for details.
>
> >**Q5**: Does the rank saturation phenomenon also happen in real datasets when the input dimension n varies?
>
> **A5**: Please refer to **A3** for details.

---

> > ### Comment · Reviewer_VcLz · 2024-11-29
> >
> > I thank the authors for the response and the revised manuscript. The additional experiment and the new proof with the perturbation analysis greatly improved the paper and I now vote for acceptance.
> >
> > In case the paper doesn't get accepted, I encourage the authors to resubmit and I have some suggestions for improving the presentation that I think could benefit the paper:
> >
> > 1) It would be better to present theorem 1 already using the weaker assumption about the data. Also, the allowed size of the perturbation is only presented implicitly (Appendix B.1, Lemma 6). It can be made explicit as part of the theorem's assumptions.
> >
> > 2) Figures 15 and 16 are very good, I think they should be part of the main paper rather than the last page of the appendix. Also, the rank (if I understand correctly) is calculated only for the first layer. It can be beneficial to have another figure representing the rank saturation phenomenon at different layers.
> >
> > 3) The abstract and intro are still a bit misleading since it is not clear from them that the theoretical analysis is about random weights. I understand this is discussed in Section 5, but it would be fair to say this much earlier in the paper.

---

> > > ### Author Response · Authors · 2024-12-02
> > > **Response to Reviewer VcLz**
> > >
> > > We are very glad to see that our revisions solve most of your questions, and many thanks for you to raise the score.
> > >
> > > For your follow-up feedback, we have also adopted all of them in the new section **Newly updated results (continue)** in the global response (since a pdf revision cannot be uploaded currently, we list the changes there to inform all reviewers). Again, we are very grateful to have these insightful suggestions from you, which greatly help us to improve this work.

---

### Official Review · Reviewer_CvzB · 2024-11-02

**Soundness:** 2
**Presentation:** 2
**Contribution:** 2
**Rating:** 3
**Confidence:** 3

**Summary:**

The manuscript investigates the relationship between head dimensions and the corresponding attention score matrix ranks. The authors identify two phenomena in Transformer attention ranks: (1) an upper bound on attention rank, referred to as the "low-rank barrier," and (2) the "model-reduction effect," which denotes the diminishing returns on attention rank when increasing head dimensions beyond a certain threshold. Experiments are provided to validate these findings.

**Strengths:**

1. The paper offers a new theoretical perspective on Transformer architecture, particularly in characterizing the limitations of attention ranks, which is insightful for understanding the underlying mechanics of model capacity.

2. The two phenomena in Transformer attention ranks: (1) an upper bound on attention rank, referred to as the "low-rank barrier," and (2) the "model-reduction effect," which denotes the diminishing returns on attention rank when increasing head dimensions beyond a certain threshold, are quite interesting and intriguing.

**Weaknesses:**

1. Lack of motivation: First of all, why should one care about the rank of the attention matrix? While it is interesting to note that the attention matrices display the low-rank barrier and model-reduction effects, it is unclear how these findings directly impact the design or usage of Transformer models in practical applications. The study would benefit from a more explicit motivation linking these theoretical insights to specific challenges in machine learning or computational limitations. In particular, attention ranks do not seem to have a clear relationship with the model performance or expressive power. Have you identified whether the low-rank barrier correlates with any performance metrics? Could the model-reduction effect be leveraged to improve model efficiency?

2. Assumptions in theoretical analysis: The theoretical analysis assumes orthonormal input sequences to attention, which may not fully reflect the reality. For example, there are other evidence in the literature suggesting that contexualized token embeddings tend to be anisotropic [Ethayarajh, 2019]. While the authors justify this orthogonal assumption by citing Tian et al. 2024, further discussion on the applicability of the theoretical results in varied real-world scenarios would enhance their robustness.

3. Limited exploration of practical applications: While the theoretical findings are interesting, the work could benefit from a more explicit discussion of how these insights translate into practice. For example, I would be interested to see if the findings on the model-reduction effect could lead to model compression techniques without significant performance loss.



Kawin Ethayarajh. How contextual are contextualized word representations? comparing the
geometry of bert, elmo, and gpt-2 embeddings. In EMNLP, 2019.

**Questions:**

See above.

---

> ### Author Response · Authors · 2024-11-27
> **Response to Reviewer CvzB**
>
> >**Q1**: Lack of motivation: First of all, why should one care about the rank of the attention matrix? While it is interesting to note that the attention matrices display the low-rank barrier and model-reduction effects, it is unclear how these findings directly impact the design or usage of Transformer models in practical applications. The study would benefit from a more explicit motivation linking these theoretical insights to specific challenges in machine learning or computational limitations. In particular, attention ranks do not seem to have a clear relationship with the model performance or expressive power. Have you identified whether the low-rank barrier correlates with any performance metrics? Could the model-reduction effect be leveraged to improve model efficiency?
>
> **A1**: For the rank motivation and connections with pratice:
> - Matrix rank is a fundamental algebra concept,  which reflects the spectrum of certain operators. Its fundamental effect in approximation (expressive power) has been discussed in the paragraph at Line 354-365 in the original manuscript.
> - As is discussed in the introduction part of the original manuscript (Line 45-50) , an important phenomenon called the *low-rank bottleneck* has been uncovered by numerous recent works for practical Transformers. Representative examples among them include (but not limited to)
>     - [1]: The low-rank attention unit *cannot represent* certain desired contexts.
>     - [2]: Pure attention loses its rank exponentially fast in depth, leading to rank-1 attention matrices and quite limited expressive power (outputs stay in only one dimension).
>
> For the model-reduction effect and performance metrics:
> - In the first paragraph of Section 5 (Line 422-427) in the original manuscript, we demonstrate through real-world numerical simulations that "the low-rank saturation of every single head leads to an *inefficiency* issue: Both the attention rank and model performance *consistently* get *marginal enhancements* when increasing parameters, implying the model redundancy".
> - To boost the model efficiency, this aligned saturation in both model performance and attention ranks "gives chances for the optimal configuration of hyper-parameters: In practical applications, one may check the saturation situation of attention ranks before training, and set the optimal number of parameters as where the rank first gets saturated".
>
> >**Q2**: Assumptions in theoretical analysis: The theoretical analysis assumes orthonormal input sequences to attention, which may not fully reflect the reality. For example, there are other evidence in the literature suggesting that contexualized token embeddings tend to be anisotropic [Ethayarajh, 2019]. While the authors justify this orthogonal assumption by citing Tian et al. 2024, further discussion on the applicability of the theoretical results in varied real-world scenarios would enhance their robustness.
> Kawin Ethayarajh. How contextual are contextualized word representations? comparing the geometry of bert, elmo, and gpt-2 embeddings. In EMNLP, 2019.
>
> **A2**: We argue that assuming orthonormal inputs in the main theorem gives the worst-case justification. To see this, let $\mathbf{A}:=\mathbf{X} \mathbf{W} _ q \mathbf{W} _ k^\top \mathbf{X}^\top$, it is straightforward to show that orthonormal $\mathbf{X}$ leads to the maximal $\mathrm{rank}(\mathbf{A})$. In fact, since the matrix multiplication cannot increase the overall rank, we formally have $\mathrm{rank} _ {\text{low-rank}~\mathbf{X}}(\mathbf{A}) \le \mathrm{rank} _ {\text{full-rank} ~ \mathbf{X}}(\mathbf{A})$. Also, for any full-rank matrix $\mathbf{X}$, it has a QR decomposition $\mathbf{X}=\mathbf{Q}\mathbf{R}$, where $\mathbf{Q}$ is orthonormal and $\mathbf{R}$ is a upper triangular matrix with positive diagonal elements (and hence invertible). This leads to an equivalent parameterization $\mathbf{A}=\mathbf{Q} (\mathbf{R} \mathbf{W}_q) (\mathbf{R}\mathbf{W}_k)^\top \mathbf{Q}^\top$.

---

> ### Author Response · Authors · 2024-11-27
> **Response to Reviewer CvzB (continue)**
>
> >**Q3**: Limited exploration of practical applications: While the theoretical findings are interesting, the work could benefit from a more explicit discussion of how these insights translate into practice. For example, I would be interested to see if the findings on the model-reduction effect could lead to model compression techniques without significant performance loss.
>
> **A3**: We respond by two points:
> - In the first paragraph of Section 5 (Line 425-427) in the original manuscript, we have briefly discussed the principle of hyper-parameters configuration: "In practical applications, one may check the saturation situation of attention ranks before training, and set the optimal number of parameters as where the rank first gets saturated".
> - Although it is beyond the scope of this work, we are inspired by Reviewer 64Hy for potential developments of this work in model compression techniques. Related references (e.g. KDEformer [3], HyperAttention [4]) aim to study the approximate calculation problem of attention matrices (with direct applications in model compression), with the fundamental approach to reduce the full matrix multiplication to sub-matrix multiplications. These works relate to attention ranks through the size of sub-matrices, which is typically lower bounded by measures depending on (stable) ranks of attention matrices. It would be interesting to further develop these works with the inductive biases established in this work, i.e. explore potentially more efficient algorithms given the  low-rank barrier and rank saturation of attention matrices. We have added these references and corresponding discussions in the related work section (***Section A***) in the revised manuscript.
>
>
> **References**
>
> [1] Srinadh Bhojanapalli, Chulhee Yun, Ankit Singh Rawat, Sashank Reddi, and Sanjiv Kumar. Low-rank bottleneck in multi-head attention models. *International Conference on Machine Learning*, pp. 864–873. PMLR, 2020.
>
> [2] Yihe Dong, Jean-Baptiste Cordonnier, and Andreas Loukas. Attention is not all you need: Pure attention loses rank doubly exponentially with depth. *International Conference on Machine Learning*, pp. 2793–2803. PMLR, 2021.
>
> [3] Amir Zandieh, Insu Han, Majid Daliri, and Amin Karbasi. KDEformer: Accelerating transformers via kernel density estimation. *International Conference on Machine Learning*, pp. 40605–40623. PMLR, 2023.
>
> [4] Insu Han, Rajesh Jayaram, Amin Karbasi, Vahab Mirrokni, David P. Woodruff, and Amir Zandieh. HyperAttention: Long-context attention in near-linear time. In *International Conference on Learning Representations*, 2024.

---

### Official Review · Reviewer_64Hy · 2024-11-04

**Soundness:** 1
**Presentation:** 2
**Contribution:** 1
**Rating:** 3
**Confidence:** 4

**Summary:**

This paper focuses on the attention matrix in transformers, and studies the effect of the head dimension on the rank of this attention matrix. Under assumptions on the data and the transformer weights, the paper empirically and theoretically highlight that the rank of the $(n \times n)$ attention matrix for $n$-length input sequences is upper-bounded by a quantities close to $0.63n$, and that attention matrix rank grows with the head dimension $d_h$ but the gain in the attention matrix rank diminishes as the head dimension grows, demonstrating a "diminishing returns" behaviour. This behaviour is demonstrated with vision transformers where the ranks of the attention matrix of the first transformer block are reported as the head dimensions is varied.

**Strengths:**

This paper considers an interesting topic of analysis, studying the rank of the attention matrix, and how it is limited from above, which puts a limitation on the expressivity of the attention mechanism.

**Weaknesses:**

- (W1) In my opinion, the main weakness of this paper are the analysis tools and assumptions utilized for providing the rank upper bounds which is used make the main claim of this paper that the attention matrices are rank limited. Here are some specific issues:
  - (W1.1) There are many variations of multi-head where the value matrix $\mathbf{V}^{(i)} \in \mathbb{R}^{n \times d_h}$ has the head dimension $d_h$. But usually the key and query matrices do not need to have the same dimensionality as the value matrix. Furthermore, there are versions where $d_h \times h \not= d_{\text{model}}$, and $W_o$ projects $h \times d_h \to d_{\text{model}}$. For example $d_h = d_{\text{model}}$. How does the results studied here be affected by these different strategies? Furthermore, there are versions that just vary the value matrix dimension, but the query/key matrix dimensions are not affected by the number of heads. In that case, this analysis is not applicable. In fact, in the empirical evaluation of Figure 4b (which uses a different variation), we are able to go beyond the $0.63$ range with $d_h < 5$ and goes up to 0.68. Different variations of this could allow us to go to full rank (or close to it). In fact, it would seem that, with $h = 1$, we should be able to recover the full-rank. Can the authors please explicitly discuss how their analysis might change for these different variations? Alternately, can the authors please justify their choice of focusing on one particular formulation of multi-head attention as this would clarify the scope and limitations of the presented results?
  - (W1.2) It is not clear how much of this analysis is dependent on the normal distribution assumption on the weigthts and tokens? Consider a case where the $\mathbf{K} = \mathbf{Q} = \mathbf{X}$ with each $\|\|\mathbf{x}_i \|\|_2 = 1$ and the temperature is low enough that we are doing top-1 attention (which is also what is considered in this paper), then the matrix $\textbf{Attn}(\mathbf{X}) = I_n$ which is the $n\times n$ identity matrix, which is full-rank. So this clearly gives a case where the proposed bound is violated. Why is this an implausible case, or conversely, how does the presented analysis subsume this situation? While the identity attention matrix seems too special, one can think of a problem where each token needs to just attend to the token right before it (that is $\textbf{Attn}(\mathbf{X})[i, i-1] = 1, \forall i > 1$, leading to an off-diagonal almost full-rank attention matrix. Can the authors please address this specific counterexample and explain how it relates to their assumptions and results?
  - (W1.3) It is odd that while we are studying the effect of the head dimension on the rank of the attention matrix, Theorem 1 has no dependence on $d_h$. This makes the result somewhat odd. I think this is an artifact of the assumptions and analysis, which effectively reduces the attention matrix to the case where each row has 1 in one of the $n$ indices at random in a query independent way. This is equivalent to each token sampling uniformly at random with replacement a token out of the $n$ tokens. Thus the expected number of unique tokens attended to in the complete attention matrix (with only one non-zero per row) is equivalent to its rank. Using standard combinatorics arguments, this expected number of unique tokens attended (for sampling with replacement) to will come to $n(1 - (1 - 1/n)^n)$ which approaches the $\approx 0.63n$ bound. While the analysis in the paper is correct, this form of an attention matrix is not useful or interesting for real applications of transformers, and also the head dimension $d_h$ plays no role here, which is different from the motivation of this paper. Can the authors please explicitly discuss this apparent discrepancy and explain how it relates to their overall claims about the effect of head dimension on attention rank? Alternately, the authors can also share (or point to) a finer-grained analysis that directly tie this rank upper bound to the head dimension.
  - (W1.4) It is not clear why line 363 "Recall that the rows of $\mathbf{X} \mathbf{W}_q \mathbf{W}_k^\top \mathbf{X}^\top = \mathbf{Q} \mathbf{K}^\top$ are independently and identically distributed as $\mathcal{N}(\mathbf{0}_n, \mathbf{K} \mathbf{K}^\top)$" is true. Why is this distribution independent of $\mathbf{Q}$? Similarly, equation (6) seems odd, highlighting that the rows for each query in the attention matrix are distributed identically. This query-independence is both odd and counter to the main motivation of transformers that usually have different attention patterns for different rows/queries.
- (W2) This is smaller weakness, but it is not clear what we can do with this rank-bounded-ness insight (assuming that this upper bound is useful and accurate). It is not clear what problems a transformer is unable to solve because of this rank-boundness, or what problems it would have been able to solve if it was able to have full-rank attention matrices. Can the authors please provide specific examples or hypothetical scenarios where this rank-boundedness might impact transformer performance or capabilities as this would help connect their theoretical results to practical applications.

Minor comments:
- (C1) It would be good to make the caption of Figure 1 self-sufficient (or point to the part of the text where the complete description is available). Otherwise, having such an introductory figure on page 2 seems a bit confusing.
- (C2) The "less than or around 0.01" comment in Table 2 caption makes that "log-dependence" argument a bit less convincing. One can alternately argue that for sequence length of 200, we needed more than a linear increase in $d_h$, implying a very different story.
- (C3) While the assumptions on the input are discussed in Remarks 2 and 3, note that the assumptions on the key/query projection matrices seem more restrictive to me, and require appropriate discussion.

**Questions:**

- (Q) Both KDEFormer [A] and Hyperattention [B] seems to also consider the rank of the attention matrix theoretically (among others). However, these references are missing. How is the setup of this current paper positioned against these references?
  - [A] Zandieh, Amir, et al. ["Kdeformer: Accelerating transformers via kernel density estimation."](https://proceedings.mlr.press/v202/zandieh23a.html) International Conference on Machine Learning. PMLR, 2023.
  - [B] Han, Insu, et al. ["HyperAttention: Long-context Attention in Near-Linear Time."](https://openreview.net/forum?id=Eh0Od2BJIM) The Twelfth International Conference on Learning Representations. 2024
- (Q) It is not clear how the rank of the attention matrix is tied to the expressivity of the attention mechanism. Are there existing results that make this connection?
- (Q) Given the use of softmax operation in the attention matrix, isn't it expected that the attention matrix will not be full rank? Part of the motivation for schemes like Performers [C], Scatterbrain [2] etc is this low rank structure. In fact, if we remove the softmax, this linear attention can probably have full rank if the head dimension is large enough but is not desired since the softmax operation is what makes attention work.
  - [C] Choromanski, Krzysztof Marcin, et al. ["Rethinking Attention with Performers."](https://openreview.net/forum?id=Ua6zuk0WRH) International Conference on Learning Representations. 2020.
- (Q) In Section 3.1, whose rank are we computing? The $\textbf{Attn}^{(i)}(\mathbf(X))$ matrices? How are the ranks aggregated across the multiple heads?

**Details Of Ethics Concerns:**

There are no ethical concerns in my opinion.

---

> ### Author Response · Authors · 2024-11-27
> **Response to Reviewer 64Hy**
>
> >**Q1**: There are many variations of multi-head where the value matrix $\mathbf{V}^{(i)}\in\mathbb{R}^{n \times d_h}$ has the head dimension $d_h$. But usually the key and query matrices do not need to have the same dimensionality as the value matrix. Furthermore, there are versions where $d _ h \times h \ne d _ {\text{model}}$, and $\mathbf{W} _ o$ projects $h \times d _ h \to d_{\text{model}}$. For example $d_h = d_{\text{model}}$. How does the results studied here be affected by these different strategies? Furthermore, there are versions that just vary the value matrix dimension, but the query/key matrix dimensions are not affected by the number of heads. In that case, this analysis is not applicable. In fact, in the empirical evaluation of Figure 4b (which uses a different variation), we are able to go beyond the $0.63$ range with $d_h<5$ and goes up to $0.68$. Different variations of this could allow us to go to full rank (or close to it). In fact, it would seem that, with $h=1$, we should be able to recover the full-rank. Can the authors please explicitly discuss how their analysis might change for these different variations? Alternately, can the authors please justify their choice of focusing on one particular formulation of multi-head attention as this would clarify the scope and limitations of the presented results?
>
> **A1**: Thanks for your detailed comments.
> - For variations: We would like to clarify that we have investigated both cases of single head (Theorem 1) and multiple heads (Figure 3: $d=d_{\text{model}} = h \times d_h$; Figure 4: $d \ne d_{\text{model}} = h \times d_h$). There are certainly many other variants, but we select these variations with specific purposes:
>     - Single head: It is necessary to start with the most fundamental case.
>     - Multiple heads with $d=d_{\text{model}} = h \times d_h$: It is the most standard and conventional case in practice.
>     - Multiple heads with $d \ne d_{\text{model}} = h \times d_h$: This is required by our goal to study the model reduction effect of Transformers, i.e. to check the architectural redundancy when increasing modeling parameters. Since the scope of this work is on the head dimension $d_h$, we have to vary $d_h$ with other hyper-parameters fixed.
> - Beyond the range: Note that the $\approx 0.63n$ upper bound (Eq. (4)) is derived under idealizations, and can only act as approximations of real-world experiments. This is understandable due to the complexity of ground truth distributions (e.g. images) and large-scale models (with many different modules). Moreover, as is discussed in the first paragraph of Section 5 (Line 421-426) in the manuscript, the primary goal in real-world settings is to investigate the model redundancy issue.
>
> >**Q2**: It is not clear how much of this analysis is dependent on the normal distribution assumption on the weigthts and tokens? Consider a case where the $\mathbf{K}=\mathbf{Q}=\mathbf{X}$ with each $\parallel\mathbf{x}_i\parallel_2=1$ and the temperature is low enough that we are doing top-1 attention (which is also what is considered in this paper), then the matrix $\mathbf{\text{Attn}}(\mathbf{X})=\mathbf{I}_n$ which is the $n \times n$ identity matrix, which is full-rank. So this clearly gives a case where the proposed bound is violated. Why is this an implausible case, or conversely, how does the presented analysis subsume this situation? While the identity attention matrix seems too special, one can think of a problem where each token needs to just attend to the token right before it (that is $\mathbf{\text{Attn}}(\mathbf{X})[i,i-1]=1$, $\forall i >1$, leading to an off-diagonal almost full-rank attention matrix. Can the authors please address this specific counterexample and explain how it relates to their assumptions and results?
>
> **A2**: Thanks for your detailed comments. In fact, our bound in Theorem 1 (Eq. (4)) is for $\mathbb{E} _ {\mathbf{W}}$, not for $\sup _ {\mathbf{W}}$ (which is the case you refer to), and bounded $\mathbb{E}\mathbf{z}$ does not imply the boundness of $\sup\mathbf{z}$ ($\mathbf{z}$: random variable). One can easily construct a counterexample even in one dimension: Let $z \sim p(z)=2/z^3$, $z\ge 1$, it is straightforward to verify that $p(\cdot)>0$ and $\int _ 1^\infty p(z) dz=1$ (i.e., $p(\cdot)$ is a probability density), $\mathbb{E} z=\int_1^\infty zp(z) dz=2<\infty$, while $\sup z=\infty$.

---

> ### Author Response · Authors · 2024-11-27
> **Response to Reviewer 64Hy (continue)**
>
> >**Q3**: It is odd that while we are studying the effect of the head dimension on the rank of the attention matrix, Theorem 1 has no dependence on $d_h$. This makes the result somewhat odd. I think this is an artifact of the assumptions and analysis, which effectively reduces the attention matrix to the case where each row has 1 in one of the $n$ indices at random in a query independent way. This is equivalent to each token sampling uniformly at random with replacement a token out of the $n$ tokens. Thus the expected number of unique tokens attended to in the complete attention matrix (with only one non-zero per row) is equivalent to its rank. Using standard combinatorics arguments, this expected number of unique tokens attended (for sampling with replacement) to will come to $n(1-(1-1/n)^n)$ which approaches the $\approx 0.63n$ bound. While the analysis in the paper is correct, this form of an attention matrix is not useful or interesting for real applications of transformers, and also the head dimension $d_h$ plays no role here, which is different from the motivation of this paper. Can the authors please explicitly discuss this apparent discrepancy and explain how it relates to their overall claims about the effect of head dimension on attention rank? Alternately, the authors can also share (or point to) a finer-grained analysis that directly tie this rank upper bound to the head dimension.
>
> **A3**: We respond by two points:
> - For independence on $d_h$: This is exactly what the low rank barrier stands for. In fact, Theorem 1 holds *universally* for any head dimensions. The $\approx 0.63n$ upper bound is derived under certain idealizations, but both the argument and applicability in practice are not limited, as is explained in the following points.
> - For combinatorics arguments: The standard combinatorics argument is over-simplistic, since it is based on assumptions that (i) the (column) values in attention matrices are identically distributed (leading to the $1/n$ probability to assign $1$); (ii) the rows of attention matrices are independent and identically distributed (leading to the $n$-power), and combining (i) (ii) gives the $n(1-(1-1/n)^n)$ quantity. However, the assumptions (i)(ii) are too strong with the trivial symmetry of indices. In fact, the considered setting is not trivial to tackle. The non-triviality of the current setting is, here we do *not* assume any independence in the attention matrix. Its columns are *dependent*. Although its rows can be proved to be independent due to Lemma 4 and the arguments in Line 859-866 in the original manuscript under the orthonormal inputs assumption, the orthonormality can be relaxed, as is quantitatively analyzed in newly updated Section B.1, hence no independence of rows can be guaranteed. This definitely raises more technical difficulties, which significantly distinguishes the proof framework from simple symmetry or combinatorics arguments.
>     - Also note that it is not straightforward to derive $n(1-(1-1/n)^n) \to n (1-1/e)$ as $n \to \infty$, since the multiplier $n$ also goes to positive infinity when applied the limit $(1-1/n)^n \to 1/e$ (i.e., the limit is indeterminate). As a technical solution, Lemma 2 in the original manuscript gives the required quantitative characterization.
>
> >**Q4**: It is not clear why line 363 "Recall that the rows of $\mathbf{X} \mathbf{W}_q \mathbf{W}_k^{\top} \mathbf{X}^{\top} = \mathbf{Q} \mathbf{K}^{\top}$ are independently and identically distributed as $\mathcal{N}(0_n, \mathbf{K}\mathbf{K}^{\top})$" is true. Why is this distribution independent of $\mathbf{Q}$? Similarly, equation (6) seems odd, highlighting that the rows for each query in the attention matrix are distributed identically. This query-independence is both odd and counter to the main motivation of transformers that usually have different attention patterns for different rows/queries.
>
> **A4**: The claim holds due to Lemma 4 and the arguments in Line 859-866 in the original manuscript under the orthonormal inputs assumption. Again, the orthonormality can be relaxed, as is quantitatively analyzed in newly updated Section B.1, hence no independence of rows in attention matrices can be guaranteed. Since the columns of attention matrices are also not required to be independent, our assumption does not lose generality even compared with practical applications, where the parameters and data in attention matrices are coupled after training and certainly with dependent entries.

---

> ### Author Response · Authors · 2024-11-27
> **Response to Reviewer 64Hy (continue)**
>
> >**Q5**: This is smaller weakness, but it is not clear what we can do with this rank-bounded-ness insight (assuming that this upper bound is useful and accurate). It is not clear what problems a transformer is unable to solve because of this rank-boundness, or what problems it would have been able to solve if it was able to have full-rank attention matrices. Can the authors please provide specific examples or hypothetical scenarios where this rank-boundedness might impact transformer performance or capabilities as this would help connect their theoretical results to practical applications.
>
> **A5**: Yes, as is discussed in Line 45-50 in the original manuscript, an important phenomenon called the *low-rank bottleneck* has been uncovered by numerous recent works. Representative examples among them include (but not limited to)
> - [1]: The low-rank attention unit cannot represent certain desired contexts.
> - [2]: Pure attention loses its rank exponentially fast in depth.
>
> >**Q6**: It would be good to make the caption of Figure 1 self-sufficient (or point to the part of the text where the complete description is available). Otherwise, having such an introductory figure on page 2 seems a bit confusing.
>
> **A6**: Thanks for your suggestion. We have updated this in the revised version.
>
> >**Q7**: The "less than or around 0.01" comment in Table 2 caption makes that "log-dependence" argument a bit less convincing. One can alternately argue that for sequence length of 200, we needed more than a linear increase in $d_h$, implying a very different story.
>
> **A7**: Certainly, noises are inevitable in the evaluation (even with averaging), since both the data generation and parameters initialization introduce randomness. That is why the standard deviation ($\pm$) is also reported, and a not strict tolerance ("around" $0.01$) is required. The story is not different, since one can observe (i) minor variations around the saturation; (ii) increases of $d_h^*$ (around the saturation) are much slower than those of $n$.

---

> ### Author Response · Authors · 2024-11-27
> **Response to Reviewer 64Hy (continue)**
>
> >**Q8**: While the assumptions on the input are discussed in Remarks 2 and 3, note that the assumptions on the key/query projection matrices seem more restrictive to me, and require appropriate discussion.
>
> **A8**: It is true that the main theorem holds for Gaussian distributions of model weights, which can be certainly viewed as an initialization regime. However, this formulation can be also understood as a "capacity" (expressive ability) study of Transformers, but only analyzed for random parameters. In fact, random neural networks are *commonly* studied in theoretical papers even for architectures that are much simpler than Transformers, such as random feature models, neural tangent kernels and reservoir computing.
>
> >**Q9**: Both KDEFormer [A] and Hyperattention [B] seems to also consider the rank of the attention matrix theoretically (among others). However, these references are missing. How is the setup of this current paper positioned against these references?
> [A] Zandieh, Amir, et al. "Kdeformer: Accelerating transformers via kernel density estimation." International Conference on Machine Learning. PMLR, 2023.
> [B] Han, Insu, et al. "HyperAttention: Long-context Attention in Near-Linear Time." The Twelfth International Conference on Learning Representations. 2024.
>
> **A9**: It seems that both KDEFormer [A] and Hyperattention [B] aim to study the approximate calculation problem of attention matrices, with the fundamental approach to reduce the full matrix multiplication to sub-matrix multiplications. These works relate to attention ranks through the size of sub-matrices, which is typically lower bounded by measures depending on (stable) ranks of attention matrices. It would be interesting to further develop these works with the inductive biases established in this work, i.e. explore potentially more efficient algorithms given the  low-rank barrier and rank saturation of attention matrices. We have added these references and corresponding discussions in the related work section (***Section A***) in the revised manuscript.
>
> >**Q10**: It is not clear how the rank of the attention matrix is tied to the expressivity of the attention mechanism. Are there existing results that make this connection?
>
> **A10**: Please refer to **A5** for details.
>
> >**Q11**: Given the use of softmax operation in the attention matrix, isn't it expected that the attention matrix will not be full rank? Part of the motivation for schemes like Performers [C], Scatterbrain [2] etc is this low rank structure. In fact, if we remove the softmax, this linear attention can probably have full rank if the head dimension is large enough but is not desired since the softmax operation is what makes attention work.
> [C] Choromanski, Krzysztof Marcin, et al. "Rethinking Attention with Performers." International Conference on Learning Representations. 2020.
>
> **A11**: It is true that the softmax operation makes attention work, but the single softmax does not make everything work. Besides your mentioned references involving low-rank structures, there are other issues including sparsity ([3]) and attention sinks ([4]). More importantly, one of the core contributions of this work is the saturation effect consistently occurring in both attention ranks and model performance. This point has been discussed in the first paragraph of Section 5 (Line 421-426) in the manuscript: "For the multiple heads case, we aim to emphasize the *saturation* effect via numerical simulations. That is, despite that one can increase the overall rank by concatenation, the low-rank saturation of every single head still leads to an *inefficiency* issue: Both the attention rank and model performance *consistently* get *marginal enhancements* when increasing parameters, implying the model redundancy."
>
> >**Q12**: In Section 3.1, whose rank are we computing? The $\mathbf{\text{Attn}}^{(i)}(\mathbf{X})$ matrices? How are the ranks aggregated across the multiple heads?
>
> **A12**: Yes, we compute the averaged rank over every attention head. For the discussion about concatenation, please refer to **A11** for details.
>
> **References**
>
> [1] Srinadh Bhojanapalli, Chulhee Yun, Ankit Singh Rawat, Sashank Reddi, and Sanjiv Kumar. Low-rank bottleneck in multi-head attention models. In *International Conference on Machine Learning*, pp. 864–873. PMLR, 2020.
>
> [2] Yihe Dong, Jean-Baptiste Cordonnier, and Andreas Loukas. Attention is not all you need: Pure attention loses rank doubly exponentially with depth. In *International Conference on Machine Learning*, pp. 2793–2803. PMLR, 2021.
>
> [3] Beidi Chen, Tri Dao, Eric Winsor, Zhao Song, Atri Rudra, and Christopher Ré. Scatterbrain: Unifying sparse and low-rank attention approximation. *Advances in Neural Information Processing Systems 34*, 2021.
>
> [4] Guangxuan Xiao, Yuandong Tian, Beidi Chen,  Song Han, amd Mike Lewis. Efficient streaming language models with attention sinks. In *International Conference on Learning Representations*, 2024.

---

> > ### Comment · Reviewer_64Hy · 2024-12-02
> >
> > I thank the authors for their detailed responses and the updated results and analyses. Based on these responses and the updated manuscript, this is my understanding of the main contributions of this submission:
> >
> > - (**Contrib 1**) The first contribution is that the paper empirically demonstrates the existence of a low-rank bottleneck in the attention matrices, and some relationship between the head dimension $d_h$ and the attention rank.
> >   - However, it is not clear why this is a novel contribution given the existing literature such as Bhojanapalli et al. (2020), Dong et al. (2021) and others.
> > - (**Contrib 2**) To the best of my understanding, the results in Theorem 1 (or extended Theorem 2) establish an upperbound on the rank of a single head attention matrix under assumptions of orthogonality among the token embeddings of the input sequence ($\mathbf{X} \mathbf{X}^\top = \mathbf{I}_n$) and Gaussianity of the attention parameters $\mathbf{W}_q, \mathbf{W}_k$, while Appendix B.1 relaxes the orthogonality condition to almost orthogonality. This result (Theorem 1) is utilized to present some "model-reduction effect".
> >   - I understand that assumptions are necessary to study the theoretical properties of models. Furthermore, one can potentially justify the Gaussianity and orthogonality assumptions.
> >   - However, the moment we arrive at a situation where the (pre-soft/hard-max) attention score matrix row is identically distributed regardless of the query as in equation (28), the analysis seems overly simplified. We arrive at $\mathbf{K} \mathbf{q}_i \sim \mathcal{N} ( \mathbf{0}_n, \mathbf{K} \mathbf{K}^\top )$ -- note that here $\mathbf{q}_i = \mathbf{W}_q \mathbf{x}_i$ and the $\lbrace \mathbf{x}\_i \rbrace\_{i=1}^n$ are supposed to be orthogonal to each other. So, for a set of orthogonal token embeddings, we end up in a situation where their attention score rows are identically distributed. To me this analysis does not capture the right notion of rank bottleneck.
> >   - The assumptions and the resulting behaviour from (28) is utilized to the motivate the "model reduction effect" and motivate the results in (5) and (6). However, note that under the conditions of Theorem 1, the right hand side of (5) is zero (or very close to it under approximate orthogonality) unless $i = j$. This extends to the result in (6) implying $\mathbf{e}_i \mathbf{Q} \mathbf{K}^\top / \sqrt{d_h} \sim \mathcal{N}(\mathbf{0}_n, \mathbf{I}_n)$ which means that the subsequent rank in (7), (8) does not even depend on $\mathbf{X}$. So each row in the attention score matrix is just a multivariate normal distribution. While valid, this seems like an overly simplified analysis of the dependence of the rank on the head dimension $d_h$.
> >   - Given that the expressivity of attention (compared existing sequence modeling schemes) for modeling long term dependencies comes from the fact that the attention pattern is input dependent, it seems that this analysis makes assumptions that lead to a situation where the attention pattern is input independent, and thus very different from what happens in practice.
> >   - There is no question that the attention matrix would be low-rank. This is the premise for various efficient low-rank approximation of the attention matrix. However, I do not think this analysis captures the actual behavior underlying the attention mechanism.
> >
> > I have also read the reviews posted by other reviewers who have scored this paper highly just to make sure that I have not misunderstood the contributions. However, their reviews, responses and discussions do not clarify my technical concerns.

---

> ### Author Response · Authors · 2024-12-04
> **Response to Reviewer 64Hy**
>
> Thanks for your comments. Regarding your concerns, we would like to further clarify as follows.
>
> 1. Literature comparisons
> - As we have stated in the above rebuttal, Bhojanapalli et al. (2020) shows that low-rank attention unit cannot represent certain desired contexts (with critical "phase transitions" around the *full-rank*), and Dong et al. (2021) proves that pure attention loses its rank exponentially fast in depth (i.e. converging to *1-rank*). Their settings are totally different from this work.
> - Particularly, these works, also including many other studies, did not explicitly derive **quantitative estimates** on the attention rank, and further show the architectural **redundancy** (in both attention ranks and model performance).
>
> 2. Theoretical constructions
> - We point out that the constructions you refer to are under the "limit" setting (i.e. $\epsilon=0$ with $\epsilon$ as the perturbation tolerance). However, we have extended the results to the "asymptotic" setting (i.e. $\epsilon=o(1)\ne 0$). As is discussed in **A2** in **Response to Reviewer VcLz**,  due to Johnson–Lindenstrauss lemma, almost orthonormality (the asymptotic setting) leads to *exponentially* many "dimensions", rather than the *linear* dimension for exact orthonormality (the limit setting). Therefore, this extension is significant and non-trivial.
> - Particularly, all the independence you refer to appears in the limit setting, but *none of the independence holds in the asymptotic setting* (where Lemma 4 does not hold). That is, the independence in theoretical constructions is only *intermediary*, which is leveraged by further (approximation) analysis to obtain *general results without independence*.
> - Given that it is difficult to directly analyze the general dependence in theory, we also conducted extensive experiments to verify the theoretical results. For more complex cases, say *more types of data distributions* and *non-i.i.d.* data, *even under real-world settings*, it is observed that similar patterns of both the rank barrier and rank saturation hold (see Figure 2(right), Figure 3(b), Figure 4(b), Figure 15(right) and Figure 16 in the revised manuscript), despite that the independence in theoretical constructions is not provided in these practical cases.
>
> 3. Contributions: We kindly remind you (and also other reviewers) of possibly missing core contributions of this work. **We emphasize that the main contributions of this submission also include the key ___theory-experimentation consistency___ in ___real-world___ settings**.
> - As is shown in Figure 3(b), Figure 4(b), Figure 15(right) and Figure 16 in the revised manuscript, it is observed in general that both the low-rank barrier and rank saturation appear in Transformers (with random weights) on real-world datasets.
> - **More importantly**, as is *jointly* shown in subplots ((a), (b)) of Figure 3, 4, 15, there is significant *alignment* between the attention *rank saturation* and *model performance improvement*: When increasing the head dimension, *similar trends* of *marginal enhancements* appear in *both* the attention rank (of randomly initialized Transformers) and classification accuracy (of trained practical Transformers) on real-world datasets.
>
> This theory-experimentation consistency captures the actual behavior (i.e. architectural redundancy) underlying the attention mechanism.
>
> 4. Paradigm: Given the high non-linearity of Transformers and complexity of practical data distributions, it is considerably difficult and also unnecessary to directly analyze every mathematical detail. As is often the case to handle complex systems, here we adopt a two-stage paradigm:
> - Scale-down: For complicated practical problems, approximately propose an idealized formulation, and then perform rigorous mathematical analysis.
> - Scale-up: Back to original problems, experimentally verify the "scale-down" results to further justify their validity in general cases.
>
> This paradigm is guaranteed to be reasonable if theoretical insights (1st stage) and experimental phenomena (2nd stage) are *consistent*. As is highlighted in **Point 3**, we have achieved this ultimate goal.

---

### Official Review · Reviewer_Kafv · 2024-11-04

**Soundness:** 4
**Presentation:** 4
**Contribution:** 4
**Rating:** 8
**Confidence:** 3

**Summary:**

This work explores the architectural limitations and redundancy in Transformer models by analyzing the ranks of their attention score matrices. Through extensive experiments across diverse model configurations and data distributions, the authors uncover two key properties: the low-rank barrier and the model-reduction effect. These findings are rigorously supported by a fine-grained mathematical analysis, revealing (i) a consistent theoretical upper bound on the attention rank (0.63n) and (ii) a critical threshold for rank saturation where the hidden dimension h scales as Ω(log n). These results illuminate the inductive biases and internal dynamics of Transformers, deepening our theoretical understanding and enabling better assessment of model capacity and efficiency in practical applications. These insights are particularly valuable for Transformer architecture design and optimization.

**Strengths:**

This work studies a fundamental problem in Transformer model efficiency. In this paper, the authors present extensive empirical results and rigorous theoretical analysis, offering critical insights into the architectural limitations and redundancy in Transformer models. The findings are highly valuable for designing more efficient Transformer-based architectures.

**Weaknesses:**

While the paper presents extensive experiments on various model configurations and data distributions, the evaluation focuses on a few common computer vision datasets (CIFAR-10, CIFAR-100, and SVHN). This raises the question of whether the findings generalize to other domains, such as natural language processing or audio processing, where Transformer models are widely used. Including additional experimental results from these domains would be very helpful.

**Questions:**

The findings are rather interesting. Would they apply to other Transformer models, such as those used in NLP and audio processing tasks?

---

> ### Author Response · Authors · 2024-11-27
> **Response to Reviewer Kafv**
>
> >**Q1**: While the paper presents extensive experiments on various model configurations and data distributions, the evaluation focuses on a few common computer vision datasets (CIFAR-10, CIFAR-100, and SVHN). This raises the question of whether the findings generalize to other domains, such as natural language processing or audio processing, where Transformer models are widely used. Including additional experimental results from these domains would be very helpful.
>
> **A1**: Thanks for your suggestions. We have added required NLP experiments (on the IMDB dataset) and discussions in ***Section D.3 (Figure 15 and 16)*** in the revised manuscript. It turns out that the insights and implications obtained under image settings consistently hold under text settings with varied input sizes.
>
> >**Q2**: The findings are rather interesting. Would they apply to other Transformer models, such as those used in NLP and audio processing tasks?
>
> **A2**: Please refer to **A1** for details.

---

### Author Response · Authors · 2024-11-27
**Newly updated results (as suggested)**

We sincerely appreciate all reviewers for their insightful and constructive feedback. Besides answering questions in detail to address all  reviewers’ comments, we want to summarize and highlight new key results as follows, and all the results are included in the revised manuscript (revised contents in blue).

1. Theoretically, we have successfully extended the main theorem to the almost orthonormality setting via approximation procedures and stability/perturbation analysis. See details in ***Section B.1*** in the revised version.
2. Experimentally, we have added NLP experiments (on the IMDB dataset) and discussions in ***Section D.3 (Figure 15 and 16)*** in the revised version. It turns out that the insights and implications obtained under image settings consistently hold under text settings with varied input sizes.

---

> ### Author Response · Authors · 2024-12-02
> **Newly updated results (continue)**
>
> As is suggested by Reviewer VcLz, we further make the following revisions (since a pdf revision cannot be uploaded currently, we list the changes below):
> 1. We merge Theorem 1 and Remark 2 (Section B.1) in the revised manuscript, to form a main theorem with explicitly weaker conditions on inputs. Theorem 1 now becomes:
> Let the parameters $\mathbf{W} _ q, \mathbf{W} _ k$ be Gaussian random matrices, i.e., the entries of $\mathbf{W} _ q, \mathbf{W} _ k$ are independent $\mathcal{N}(0, 1)$ random variables. Assume that the input sequence $\mathbf{X}$ satisfies $\mathbf{X} \mathbf{X}{^\top} = \mathbf{I} _ n  +\mathbf{E}$ with $\mathbf{E} = [E _ {ij}] \in \mathbb{R} ^ {n \times n}$ satisfying $|E _ {ij}| \le \epsilon = o(1/(n ^ {\frac{3}{2}}(d+d _ h)))$  ($\forall i, j \in [n]$, i.e. almost orthonormality). Then for any $n \in \mathbb{N} _ +$ appropriately large, $d \ge n$, and $\delta>0$ appropriately small, we have
>     \begin{equation}
>         \mathbb{E} _ {\mathbf{W} _ k, \mathbf{W} _ q}
>         \left[\mathrm{rank}
>         \left(\mathrm{hardmax}
>         \left(\mathbf{X} \mathbf{W} _ q \mathbf{W} _ k^\top \mathbf{X}^\top \right), \delta
>         \right)\right]
>         \le  (1 - \exp(-1))n
>         + O(1)
>         \approx 0.63n,
>     \end{equation}
>     where $\mathrm{rank}(\mathbf{A}, \delta)$ equals to the number of  singular values (of $\mathbf{A}$) no less than $\delta$ (i.e. numerical rank).
>
> 2. We will move Section D.3 (Figure 15 and 16) to the main part, right in Section 5.2 (after Figure 4).
>
> 3. We further add an additional test on the rank saturation of different layers. The results are as follows:
>
> - Table 1: Rank saturation in layers (2nd layer). One can observe that given different (input) embedding dimensions ($32, 128, 256, 512$), the attention rank always gets saturated when increasing the head dimension.
> |Embed Dim|Head Embed Dim|Mean Rank|Std Rank
> |----|----|----|----|
> 32|2|0.0609|0.0026
> 32|3|0.127|0.0055
> 32|4|0.2032|0.0083
> 32|8|0.2539|0.0137
> 32|16|0.2584|0.0186
> 128|2|0.0591|0.0014
> 128|3|0.132|0.0037
> 128|4|0.2033|0.0063
> 128|8|0.2593|0.0096
> 128|16|0.2667|0.0101
> 256|2|0.0588|0.0018
> 256|3|0.1347|0.0051
> 256|4|0.2105|0.0074
> 256|8|0.261|0.0137
> 256|16|0.2661|0.0143
> 512|2|0.0596|0.001
> 512|3|0.134|0.0036
> 512|4|0.209|0.0061
> 512|8|0.2609|0.0106
> 512|16|0.2612|0.0108
>
> - Table 2: Rank saturation in layers (4th layer). One can observe that given different (input) embedding dimensions ($32, 128, 256, 512$), the attention rank always gets saturated when increasing the head dimension.
>   |Embed Dim|Head Dim|Mean Rank|Std Rank
> |----|----|----|----|
> 32|2|0.0518|0.0042
> 32|3|0.1177|0.0038
> 32|4|0.1925|0.0115
> 32|8|0.2584|0.0161
> 32|16|0.267|0.0137
> 128|2|0.0521|0.0023
> 128|3|0.1201|0.002
> 128|4|0.196|0.0049
> 128|8|0.2572|0.0131
> 128|16|0.2527|0.0097
> 256|2|0.0523|0.0021
> 256|3|0.121|0.0034
> 256|4|0.1953|0.0058
> 256|8|0.2677|0.0146
> 256|16|0.2551|0.0081
> 512|2|0.053|0.0021
> 512|3|0.1202|0.0056
> 512|4|0.1946|0.0066
> 512|8|0.2567|0.0134
> 512|16|0.2598|0.0139
>
> We will plot these two tables in the following version and also add them to Section 5.2 (after Figure 4).
>
> 4. To clarify the setting of theoretical analysis:
> - In the abstract, line 023-024: "we provide rigorous demonstrations for these observations through a fine-grained mathematical analysis" -> "we provide rigorous demonstrations for these observations under idealized settings through a fine-grained mathematical analysis".
> - In the introduction, line 099-100: "mathematical estimates are established on the barrier of attention ranks" -> "mathematical estimates are established on the barrier of attention ranks for Transformers with random parameters".

---

### Meta-Review · Area_Chair_GxP9 · 2024-12-20

**Metareview:**

The paper analyzes limitations of transformer architectures by studying the rank of attention score matrices, limited to the case of random data and/or random weights, showing theoretical upper bounds on attention rank and empirically demonstrating rank saturation effects as model dimensions increase.

The main concerns that remained after feedback were the theoretical analysis relying on overly simplified assumptions, such as the assumption of query-independent attention patterns, and the strong assumptions of random (not real) weights or data respectively.

We hope the detailed feedback from the reviews and discussions helps to strengthen the paper for a future occasion.

**Additional Comments On Reviewer Discussion:**

The author feedback phase was useful and active. Concerns however remained if the work is ready for the high bar of ICLR.

---

### Decision · Program_Chairs · 2025-01-22

Reject